# Diatom fucan polysaccharide precipitates carbon during algal blooms

Silvia Vidal-Melgosa [1,2], Andreas Sichert [1,2], T. Ben Francis[1], Daniel Bartosik [3,4], Jutta Niggemann[5], Antje Wichels[6], William G. T. Willats[7], Bernhard M. Fuchs [1], Hanno Teeling[1], Dörte Becher [8], Thomas Schweder [3,4], Rudolf Amann [1] & Jan-Hendrik Hehemann [1,2✉]

The formation of sinking particles in the ocean, which promote carbon sequestration into deeper water and sediments, involves algal polysaccharides acting as an adhesive, binding together molecules, cells and minerals. These as yet unidentified adhesive polysaccharides must resist degradation by bacterial enzymes or else they dissolve and particles disassemble before exporting carbon. Here, using monoclonal antibodies as analytical tools, we trace the abundance of 27 polysaccharide epitopes in dissolved and particulate organic matter during a series of diatom blooms in the North Sea, and discover a fucose-containing sulphated polysaccharide (FCSP) that resists enzymatic degradation, accumulates and aggregates. Previously only known as a macroalgal polysaccharide, we find FCSP to be secreted by several globally abundant diatom species including the genera *Chaetoceros* and *Thalassiosira*. These findings provide evidence for a novel polysaccharide candidate to contribute to carbon sequestration in the ocean.

[1] Max Planck Institute for Marine Microbiology, 28359 Bremen, Germany. [2] University of Bremen, Center for Marine Environmental Sciences, MARUM, 28359 Bremen, Germany. [3] Pharmaceutical Biotechnology, Institute of Pharmacy, University of Greifswald, 17489 Greifswald, Germany. [4] Institute of Marine Biotechnology, 17489 Greifswald, Germany. [5] University of Oldenburg, Institute for Chemistry and Biology of the Marine Environment, 26129 Oldenburg, Germany. [6] Alfred Wegener Institute, Helmholtz Center for Polar and Marine Research, Biologische Anstalt Helgoland, 27498 Helgoland, Germany. [7] Newcastle University, School of Natural and Environmental Sciences, Newcastle upon Tyne, NE1 7RU, UK. [8] Institute of Microbiology, University of Greifswald, 17489 Greifswald, Germany. ✉email: jhhehemann@marum.de

An important pathway for carbon sequestration in the ocean is the growth, aggregation and sinking of phytoplankton cells[1–3]. This so called biological carbon pump accounts for ~70% of the annual global carbon export to the deep ocean; an estimated 25–40% of carbon dioxide from fossil fuel burning emitted during the Anthropocene may have been transported by this process from the atmosphere to depths below 1000 m, where carbon can be stored for millennia[4]. During growth and upon death, diatoms release copious amounts of unknown adhesive polysaccharides, which are thought to play an important role as a carbon sink, either in high molecular weight dissolved organic matter (HMWDOM)[5] or when they become part of particulate organic matter (POM)[6]. With increasing concentrations, dissolved adhesive polysaccharides spontaneously assemble into non-covalently linked molecular networks[7]. These aggregate into small transparent exopolymer particles (TEP, > 0.4 µm), which in turn aggregate with diatom cells, minerals and other materials into larger marine snow particles (500 µm and larger), exporting carbon with a sinking rate of about 100 m/day[8–13]. To maintain the integrity of particles at this speed down to 1000 m, the polysaccharide matrix must theoretically resist biodegradation by bacterial enzymes for at least 10 days or even longer for slower sinking particles. A microalgal polysaccharide that remains stable for more than 10 days in the ocean remains unknown and intriguing considering polysaccharide-degrading bacteria and their enzymes are highly active during algal blooms[14].

Identifying such stable polysaccharides in nature requires structural information with higher molecular resolution than what is possible with currently used methods. Typically, polysaccharides are quantified in marine organic matter by chromatography of monosaccharides after acid hydrolysis[15], which destroys their native higher order structure. Alternative structural elucidation by nuclear magnetic resonance or mass-spectrometry requires purified analytes, which remains challenging for complex organic matter mixtures and water-insoluble polysaccharides[5]. Although marine particles contain up to 30 wt% of polysaccharides[16], it has yet to be tested if they are of algal, bacterial or animal origin and so the contribution of individual polysaccharides to particle formation, their structure and their stability remains unknown. Extensive bacterial activity transforming original algal polysaccharides into new ones might be an alternative source of unknown yet stable polysaccharides in the ocean[17].

Here, to identify stable adhesive glycans involved in particle formation, we traced the abundance of polysaccharide epitopes in HMWDOM and POM along a three month diatom bloom period. We used a bioanalytic approach based on carbohydrate microarrays and monoclonal antibodies (mAbs), which is widely used in medical and plant research[18–20]. mAbs are well suited for the detection of biomacromolecules owing to their high chemoselectivity and affinity. Because mAbs discriminately bind their target molecules, they can directly recognise, without prior hydrolysis or chromatography, which types of molecules, in this case polysaccharides, occur in complex marine organic matter. We found that in contrast to other detected glycans, fucose-containing sulphated polysaccharide (FCSP) accumulated for almost two months in POM and aggregated, indicating higher stability. Metaproteomics and metagenomics on bacterioplankton showed a high abundance of enzymes for the degradation of labile polysaccharides such as laminarin and low abundance of enzymes for the degradation of FCSP. The discovery of FCSP in diatoms, with demonstrated stability and adhesive properties, provides a previously uncharacterised polysaccharide that contributes to particle formation and potentially therefore to carbon sequestration in the ocean.

## Results

**Screening for polysaccharides during diatom blooms.** We monitored diatom blooms in the North Sea (54°11.3′N, 7°54.0′E) near the island of Helgoland, Germany, for about three months. Spring diatom blooms progressed from the 8th of March until the 17th of May 2016 (Fig. 1a). Chlorophyll a, a proxy for algal growth, fluctuated forming several maxima and minima ranging between 2.1 and 11.8 mg m$^{-3}$ (Supplementary Fig. 1b), provided by consecutive diatom peaks. The microalgae blooms were dominated by centric diatoms while coccolithophorids were present at lower abundances (Supplementary Fig. 2). The present study is based on samples collected from a coastal site that is comparable regarding diatom composition, abundance and productivity to many of the most productive carbon exporting regions of the ocean[21,22]. Continuous access to this site allowed us frequent sampling to assess the temporal dynamics of polysaccharides and thereof infer the reactivity of polysaccharides along an entire algal bloom period. The term bloom throughout the text refers to the series of diatom blooms covered during the studied spring bloom period.

Monosaccharide analysis did not detect temporal changes in the polysaccharide pool. POM samples were collected twice per week and separated by serial filtration into three size fractions: >10 µm, between 10 and 3 µm, and between 3 and 0.2 µm. The filtrate of the 0.2 µm filter was concentrated with tangential flow filtration on a 1 kDa membrane to obtain HMWDOM (Fig. 1b). The dissolved organic carbon (DOC) concentrations in DOM (<0.2 µm) ranged between 96 and 133 µmol L$^{-1}$ (Supplementary Fig. 1b), indicating a relatively constant amount of carbon in DOM during the bloom. To investigate if the carbohydrate composition changed or was also rather steady over time, HMWDOM samples were hydrolysed with acid into monomers and quantified by chromatography (HPAEC-PAD). Except for the relative decrease of glucosamine and a peak of arabinose at the end of the bloom, the relative abundance of monomers remained comparatively constant (Supplementary Fig. 3) and is similar in composition to previous HMWDOM results across the globe from different ocean basins, both coastal and open ocean[5].

The use of molecular probes combined with carbohydrate microarrays enabled us to detect fluctuations in specific polysaccharide structures in HMWDOM and POM. Polysaccharides from all HMWDOM and POM samples were sequentially extracted with the solvents H$_2$O, EDTA and NaOH (see Methods section), which are commonly used to release water soluble, anionic and water insoluble polysaccharides (such as laminarin, pectins, hemicelluloses and cellulose), respectively[23,24]. A library of identical microarrays was created, each populated with the same time-series of extracted polysaccharides (shown schematically in Fig. 1b). These microarrays were then individually incubated with one of 51 polysaccharide-specific mAbs or carbohydrate binding modules (CBMs) as probes, which specifically bind to a single polysaccharide epitope (probe specificities are provided in Supplementary Table 1). The binding of probes to polysaccharide epitopes on the microarrays was detected using a secondary antibody coupled to alkaline phosphatase, which converts its substrate into a coloured product, the amount of which correlates with polysaccharide concentration[20,23]. For an examination of the reproducibility, robustness and specificity of this semiquantitative method see Supplementary Discussion (Supplementary Fig. 4a, b). The microarray data set revealed the temporal dynamics of 27 polysaccharide epitopes in HMWDOM and POM (Supplementary Fig. 5). Among others, we detected β-1,3-glucan (energy storage laminarin-like structure), cell wall β-1,4-mannan, β-1,4-xylan, α-1,5-arabinan, β-1,4-galactan, homogalacturonan, cellulose, arabinogalactan and FCSP, which partitioned differently within HMWDOM and POM (data subset shown in Fig. 1c–g, complete data set in Supplementary Fig. 5). The temporal dynamics of

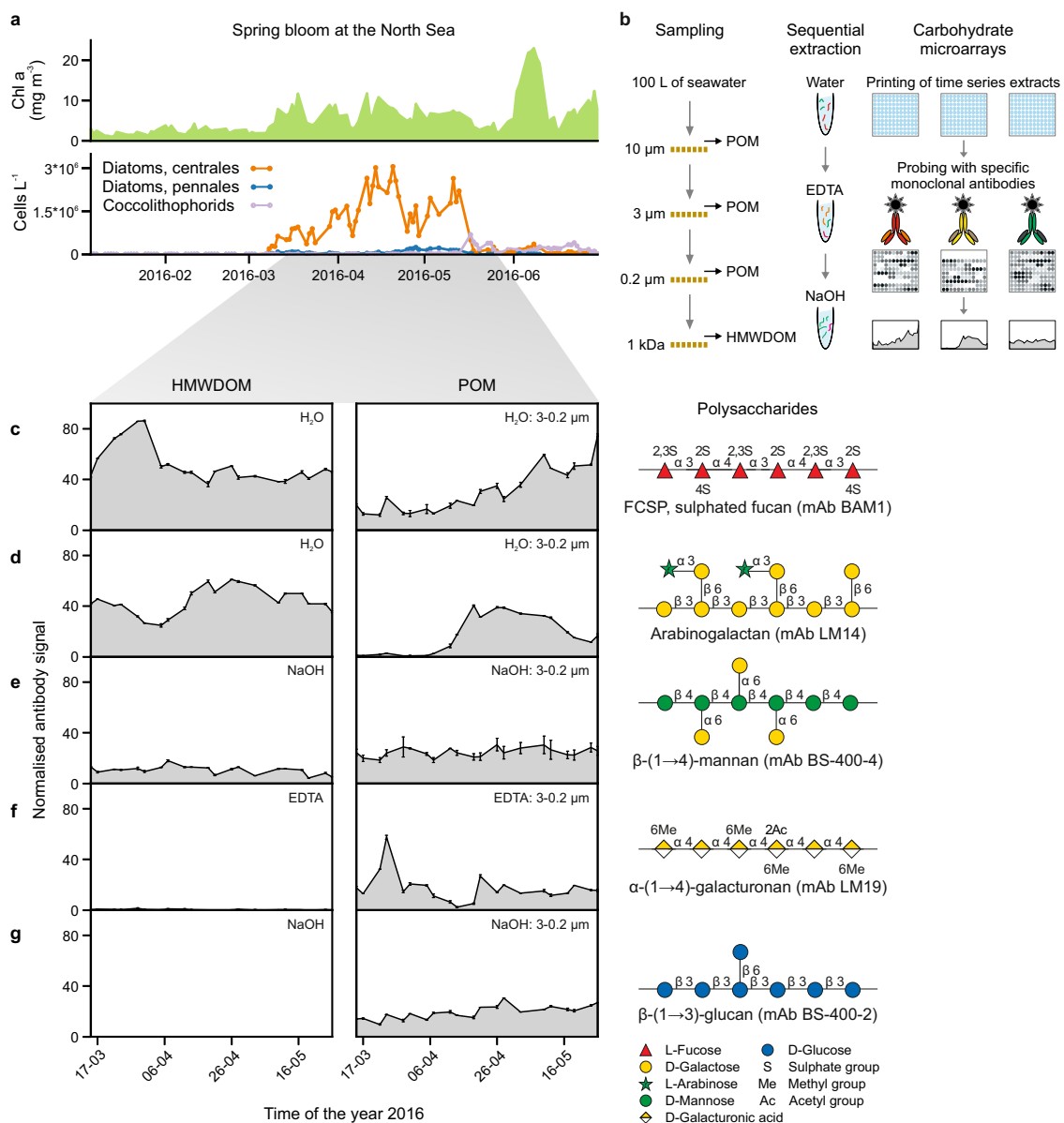

**Fig. 1 Different polysaccharide structures are present in high molecular weight dissolved organic matter (HMWDOM) and particulate organic matter (POM) and show fluctuations in their abundance during the diatom bloom. a** Chlorophyll a (Chl a) concentrations and abundances of the major microalgae taxa detected at our sampling site (54˚11.3′N, 7˚54.0′E) from January to June 2016. **b** Scheme of the sampling and carbohydrate microarray analysis. **c–g** Representative examples of five selected polysaccharides, complete microarray data set is shown in Supplementary Fig. 5. On the left, plots show the relative abundance of polysaccharide epitopes (antibody signal intensity, y axis) detected in HMWDOM and POM during the bloom (21 sampling dates, x axis). Spot signal intensities for each extract (each extract was represented by 4 spots in the array) against each probe were quantified and the highest mean signal value in the data set for HMWDOM and for POM was set to 100 and all other values were normalised accordingly. Data are mean values, $n = 4$ spots per extract. The temporal dynamics but not the absolute number should be compared between HMWDOM and POM pools as they required independent normalisation, since they required different sampling strategies. At the right, sketch of polysaccharide structures that the corresponding monoclonal antibodies (mAbs), which are depicted in parentheses, bind to. HMWDOM, between 0.2 μm and 1 kDa. Error bars in **c–g** represent ± standard deviation.

polysaccharide fluctuation, which are not perceptible using classical analytic techniques, provide new insight regarding the stability of polysaccharides.

**FCSP accumulated for weeks and aggregated**. FCSP accumulated during the bloom suggesting it is a stable polysaccharide. Algal growth with continuous presence of diatoms (Supplementary Fig. 6a), provided almost three months to detect polysaccharides that accumulated. Assuming continuous production

by algae and degradation by bacteria, increasing concentration of a polysaccharide indicates stability. As mentioned above, we define that a stable polysaccharide that is relevant for particle formation and carbon export has to resist bacterial degradation for more than 10 days[9,13]. Thus, a stable polysaccharide has to be present and increase in POM for weeks. To hold the particle together, the backbone of the adhesive polysaccharide must not be hydrolysed by endo-acting enzymes. Enzymatic modifications of sulphate or carboxyl groups that potentially provide adhesive

properties (e.g. by sulphatases) may also alter solubility. Exo-acting enzymes that cleave monomers from the end of the polysaccharide would slowly degrade it and thus slowly increase solubility. The degradation of the polymer into smaller fragments by endo-enzymes would most rapidly increase solubility[25] and therefore particle dissolution. The FCSP epitope detected with the recently developed anti-fucan mAb BAM1[26], increased for weeks in POM (Fig. 1c and Supplementary Fig. 5). Other poly-saccharides, such as the arabinogalactan and the α-1,4-galactur-onan (homogalacturonan), also increased in POM but only for days to a week (Fig. 1d, f). In contrast, some polysaccharide epitopes, such as the β-1,4-mannan recognised by mAb BS-400-4, did not accumulate (Fig. 1e). The structurally simple poly-saccharide β-1,3-glucan (laminarin-like structure), of which dia-toms produce substantial amounts[27], was close to the detection limit in HMWDOM (Fig. 1g and Supplementary Fig. 5). Minor detection of β-1,3-glucan in HMWDOM is consistent with its rapid degradation by planktonic bacteria[28]. In POM, β-1,3-glucan increased only for days to a week (Supplementary Fig. 5). The FCSP detected with mAb BAM1 was the only epitope that accumulated for weeks to month in all POM size fractions (Supplementary Fig. 5), indicating high stability, which is con-sistent with its complicated chemical structure. FCSP are α-1,3- or α-1,3;1,4-linked-L-fucose polysaccharides that have extensive sulphation along the chain and additional branching by fucose side chains and other sugars, making it difficult for microbes to degrade[29]. In addition, decreased grazing due to a potential reduced palatability of FCSP to zooplankton compared to other polysaccharides, could also contribute to its accumulation in POM. We further verified the accumulation of FCSP in POM by quantitative ELISA, and determined that there was a 3-fold increase of FCSP concentration from beginning to the end of the bloom (Supplementary Fig. 4d; discussed in Supplementary Discussion), confirming the stability of FCSP.

Theoretically, adhesive polysaccharides can transition from DOM to POM, which we only observed for FCSP detected with BAM1 (Fig. 1c and Supplementary Fig. 5). Solute decline with increase in POM indicates polymer assembly, previously demon-strated with filtered seawater where unknown anionic, dissolved polysaccharides subsequently assembled[7]. FCSP increased in March and then declined to a constant abundance from beginning of April in HMWDOM, while its presence in POM, unlike all the other detected polysaccharides, increased from April until May. This dynamic may be explained by aggregation of dissolved to particulate FCSP. This dissolved to particulate pathway is reproducible in mesocosms with diatoms and other microalgae[6,8]. Mesocosm time series experiments with cocco-lithophores showed they secrete unknown anionic, dissolved polysaccharides that aggregate into TEP[8]. TEP was also formed in mesocosms with diatoms by secreted, adhesive polysaccharides that were enriched in fucose[6]. We later tested laboratory grown cultures of diatoms (described below) and found that FCSP was abundant in the dissolved fraction, suggesting it is a secreted polysaccharide. Notably, we observed the highest increase of FCSP in the smallest particulate fraction, potentially because these smaller particles are neutrally buoyant and therefore less prone to removal by sinking. In conclusion, our data suggest that FCSP is an adhesive polysaccharide that contributes to particle formation in spring diatom blooms.

**Chaetoceros spp. are the source of FCSP**. Next, we asked which organism produced FCSP during the bloom. Using fluorescence microscopy with the mAb BAM1 in combination with a FITC-labelled secondary antibody, we visualised FCSP on filters from the bloom containing all POM sizes >0.2 μm. Fluorescence signal

was primarily localised on the surface of cells of *Chaetoceros socialis* (Fig. 2a, b), a centric, chain-forming diatom that was a dominant species continuously present throughout the bloom (Supplementary Fig. 6a). At the beginning FCSP was mainly restricted to the exterior of the cell walls and their spines (Fig. 2a). With bloom progression FCSP increasingly covered the diatoms and began appearing in particles (Fig. 2b). We quantified the FCSP signal on diatom cells and on particles (areas not con-taining diatom cells) at the beginning (March) and end (May) of the bloom. Quantification showed FCSP increasingly coated the cells (two-sided t-test, $P < 0.0001$; Fig. 2c) and revealed an increase of FCSP in particles (two-sided t-test, $P < 0.0001$; Fig. 2d). This overall increase is consistent with the FCSP increase found in the microarray POM data, which include both FCSP in diatom cells as well as in particles. Although some of the particles may derive from disrupted *C. socialis* colonies, which contain clusters of diatom chains covered in mucous that can fragment during the sampling process[30]; the observed increase of FCSP-containing particles (Fig. 2d) may be driven by aggregation of dissolved FCSP into particles. This hypothesised DOM to POM transition is consistent with the abovementioned previous mesocosm microalgae blooms where unidentified, secreted polysaccharides aggregated into TEP[8]. As TEP consist mainly of unknown, acidic polysaccharides[31], we investigated if the FCSP was negatively charged and thus could drive the formation of particles. POM and HMWDOM polysaccharides extracted with water were separated by anion exchange chromatography and all chromatographic fractions were tested by ELISA with mAb BAM1. Results showed strong binding of FCSP to the cationic column and elution (BAM1 peak) with high concentrations of NaCl, confirming FCSP is an anionic polysaccharide, both in samples from the beginning and end of the bloom (representative chromatography profiles shown in Fig. 2e).

The diatom *C. socialis* produced a negatively charged FCSP rich in fucose, galactose and sulphate. The *C. socialis* strain from the bloom was isolated during the sampling campaign; we grew laboratory monospecific cultures of this diatom, harvested the biomass, performed polysaccharide extraction and the water extract was analysed by anion exchange chromatography (Fig. 2f). Additionally, a total of 8 diatom species were analysed and presence of FCSP was confirmed in all of them (Supplementary Table 2), including two other species of the genus *Chaetoceros*, which was dominant in the bloom (Supplementary Fig. 6a). In the diatom laboratory cultures, FCSP was present in the particulate fraction, coating the diatom cells and in FCSP-containing particles, but it was also highly abundant in the dissolved fraction. This indicates FCSP is a diatom-secreted polysaccharide, part of it stays on the diatom surface coating the cell and part is released into the dissolved pool. The purified *C. socialis* FCSP (chromatographic fractions with BAM1 absorbance peak) was hydrolysed with acid into monomers, followed by quantification of released sulphate and monosacchar-ides. Our analyses confirmed the purified FCSP contained between 25 and 50% sulphate per total building blocks (Fig. 2g). The main monosaccharides of the *C. socialis* FCSP molecule were fucose, galactose, xylose and glucuronic acid, which is consistent with the main building blocks of sulphated fucans from macroalgae[29]. Also, purified FCSP from the diatoms *Thalassiosira weissflogii* (dominant genus in a previous spring bloom at the same site[14]) and *Nitzschia frustulum* (global pennate diatom) contained these four mono-saccharides (Supplementary Fig. 6b), with addition of high mannose content in *T. weissflogii* FCSP. For the FCSP purified from POM (fraction between 10 and 3 μm) and HMWDOM, the main monosaccharides were the same in both pools (Fig. 2g). Although the growth conditions differ, the composition of FCSP from seawater and from the *C. socialis* culture were similar, except for arabinose and glucose present in the FCSP from seawater (Fig. 2g

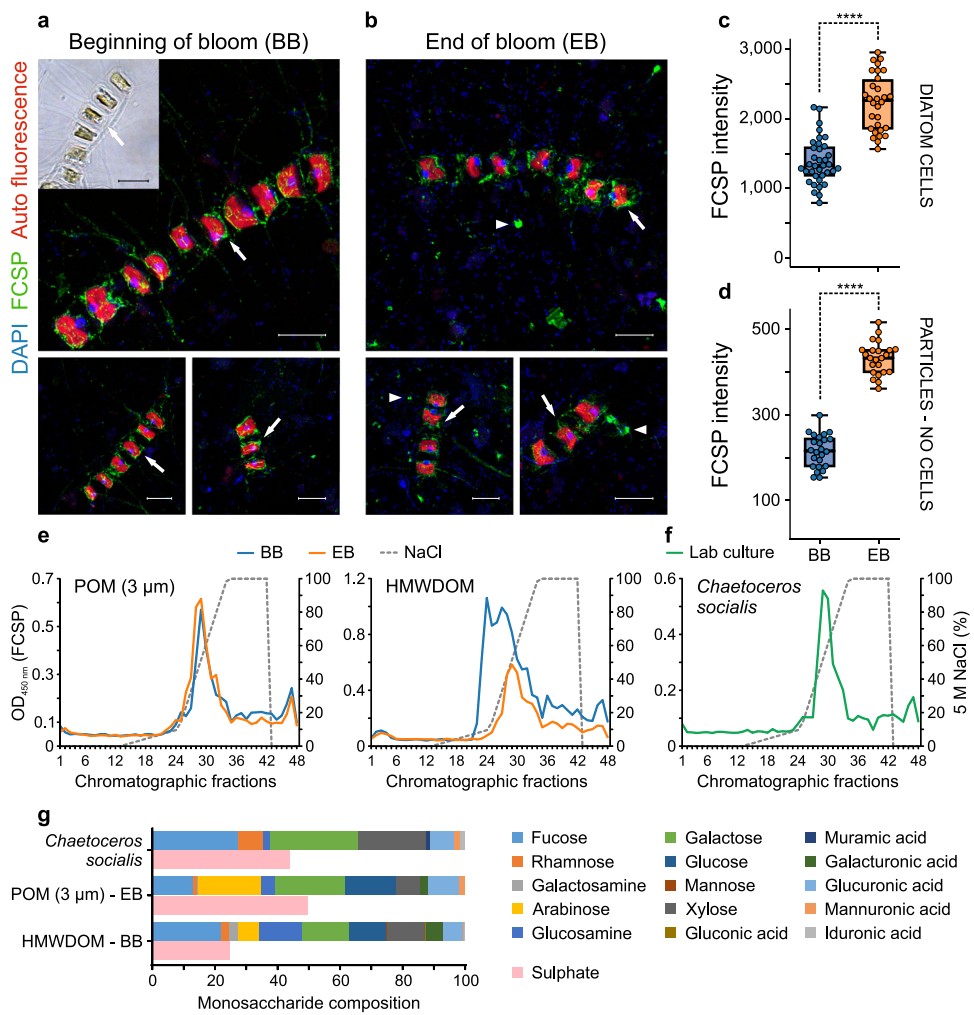

**Fig. 2 Fucose-containing sulphated polysaccharide (FCSP) is produced by diatoms and increased abundance in particulate organic matter (POM) during the bloom.** Inset in panel **a**, bright-field image of *Chaetoceros socialis* cells. **a**, **b** Representative images of FCSP localisation in POM (>0.2 μm) from the diatom bloom. Airyscan super-resolution images demonstrate FCSP occurred around the cells (arrows) of the chain-forming diatom *C. socialis* at the beginning of the bloom (BB) (**a**) and on the diatoms cells (arrows) as well as on particles (arrowheads) at the end of the bloom (EB) (**b**). In **a**, **b**, DAPI (blue), FCSP (green) and diatom auto fluorescence (red). Scale bars, 10 μm. Experiments were performed four times with similar results. **c** Quantification of FCSP signal (mAb BAM1) on diatom cells. BB $n = 36$ cells, EB $n = 30$ cells. **d** FCSP quantification on particles (areas not containing cells). BB $n = 24$ areas, EB $n = 23$ areas. In **c**, **d** ****$P < 0.0001$ (two-sided $t$-test), in **c** $P = 8.4 \times 10^{-13}$ and in **d** $P = 2.3 \times 10^{-23}$. For boxplots, the middle line indicates the median, the box designates the interquartile range (IQR) and the whiskers denote 1.5 times the IQR. **e**, **f** Chromatographic separation of FCSP in water extracts from POM and high molecular weight dissolved organic matter (HMWDOM) of beginning and end of the bloom (**e**) and from a lab culture of the diatom *C. socialis* (**f**) by anion exchange chromatography (AEC). AEC fractions were analysed by ELISA with the mAb BAM1. ELISA developing time was not the same for the five shown representative single FCSP separation AEC runs (four in **e** and one in **f**), thus absorbance values do not indicate extract concentration (see Methods section). Optical density (OD). Experiments (chromatography plus ELISA analyses) for BB and EB POM were performed two times per each, for BB and EB HMWDOM four times per each and for *C. socialis* four times, with similar results. **g** Monosaccharide composition of purified FCSP (AEC fraction with BAM1 absorbance peak) as mean relative abundance, $n = 2$ independent acid hydrolysis and HPAEC-PAD runs. See FCSP monosaccharide composition of additional AEC fractions in Supplementary Fig. 6b. Sulphate content ($n = 2$ technical replicates) of each purified FCSP sample as μM sulphate per μM total building blocks.

and Supplementary Fig. 6b). The differences in monomer composition may be a result of the higher complexity of the bloom samples with different algae and bacteria actively changing the composition of molecules in HMWDOM and POM. Overall, the molecular composition of the FCSP, its negative charge from sulphation and glucuronic acid, and its presence on algal cells and in particles, support its adhesive properties that lead to aggregation.

**Bacteria consume laminarin over FCSP.** First we analysed metagenomes of bacterioplankton to look for bacteria with genes

encoding enzymes active on β-1,3-glucan and FCSP. Notably, presence of these genes only indicates the potential to degrade these polysaccharides. Therefore, we also analysed bacterioplankton proteomes to identify if the candidate genes were also translated and expressed into enzymes. We focused on pathways for the degradation of β-1,3-glucan (laminarin-like structure) and FCSP (Fig. 3a, b) because they likely exemplify opposite ends of the lability-stability spectrum. We provide a more extensive list of metagenomic and proteomic enzymes that potentially target the other identified polysaccharides in Supplementary Table 3 and Supplementary Data 1. Microbial cell counts more than doubled

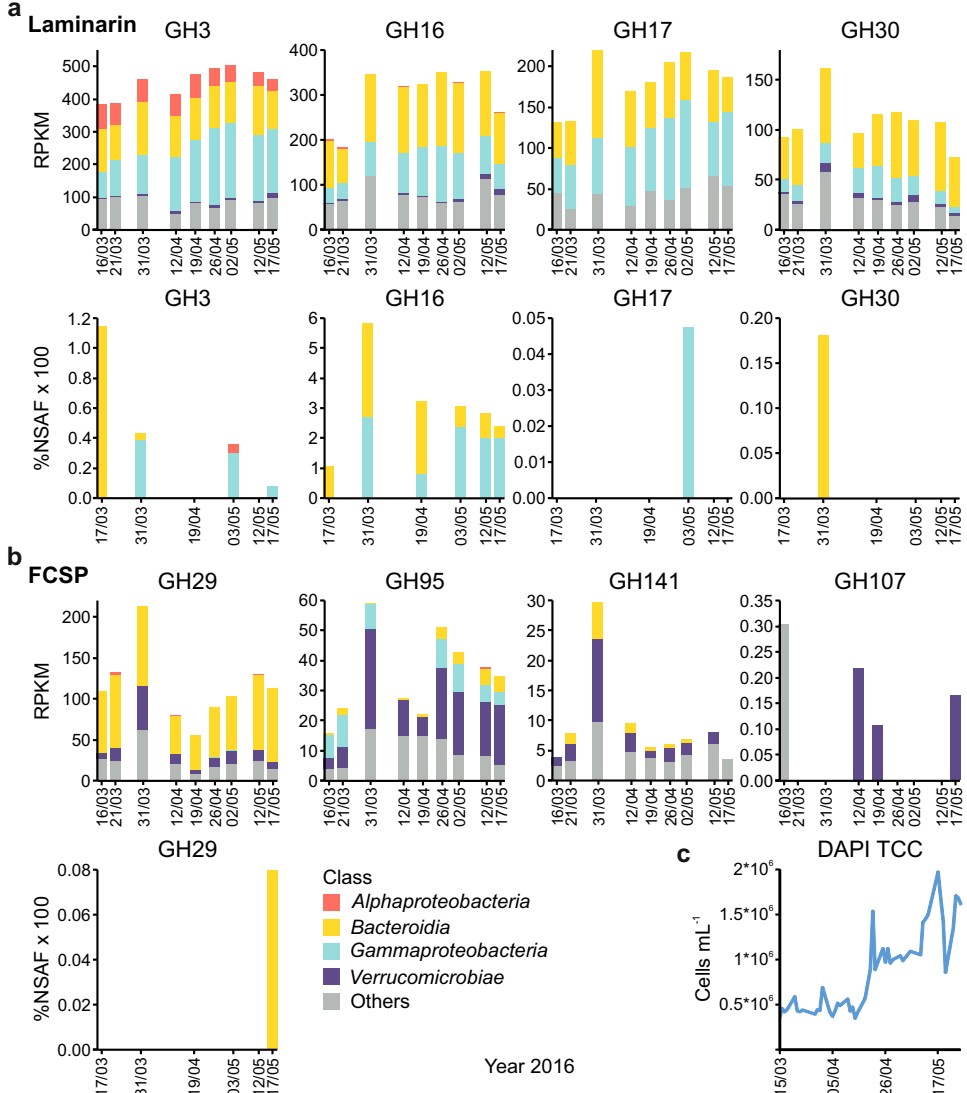

**Fig. 3 Content and expression of particular CAZymes by marine bacteria during the algal bloom.** Plots show abundances of genes coding for carbohydrate-active enzymes (CAZymes) with relevance for degradation of two selected glycan substrates in the genomes of marine bacteria (at the top of each panel) and their expression during the bloom (at the bottom of each panel). Selected substrates are: β-1,3-glucan, laminarin (**a**) and fucose-containing sulphated polysaccharide, FCSP (**b**). Complete CAZymes metagenomic analysis and complete proteomic data are shown in Supplementary Fig. 7. Reads per kilobase per million (RPKM). Proteome data were analysed in a semiquantitative manner based on normalised spectral abundance factors (%NSAF). Both analyses include class-level taxonomic classifications, see Methods section. Glycoside hydrolase family (GH). **c** DAPI-based total microbial cell counts (TCC) during the diatom bloom of 2016.

during the bloom, indicating that bacteria thrived on microalgal matter (Fig. 3c).

The metagenomic results showed that the GTDB-defined classes *Bacteroidia* and *Gammaproteobacteria* encoded enzymes for laminarin degradation (Fig. 3a and Supplementary Fig. 7a), consistent with previous results finding these groups are key degraders of laminarin[14]. Enzymes belonging to different glycoside hydrolase families (GHs) form laminarin degradation pathways that can vary in enzyme content and number. In marine bacteria, the backbone of laminarin is usually broken down by GH16 and GH17 endo-glucanases, which cleave the β-1,3-linked glucose main chain. The β-1,6 side chain glucose is removed by GH30 enzymes[32]. The resulting oligosaccharides are hydrolysed into glucose by GH3 enzymes and other glucosidases[28]. Homologs of these enzyme families were present in the metagenome data. In conclusion, the metagenomes showed the presence of bacteria that can potentially degrade laminarin with

laminarinases. Next we asked whether they expressed these enzymes.

In the metaproteomes we detected GH16 enzymes on all analysed dates. GH16 also had the highest expression levels of all the carbohydrate-degrading enzyme families within the data set (Fig. 3a and Supplementary Fig. 7b). This is consistent with GH16 laminarinases being key enzymes in laminarin degradation, frequent in laminarin-degrading bacteria[14] and in the laminarin polysaccharide utilisation loci (PULs) in the *Bacteroidetes* phylum[33]. It has been proposed that anchored outside of the cell, GH16 enzymes fragment the laminarin polysaccharide into smaller pieces for uptake into the bacterial cell; inside of the cell GH16 products are further degraded by GH3, GH30 and GH17 enzymes[28]. The proteomic data showed detection of homologs of GH3, GH30 and GH17 enzymes only on some days (Fig. 3a). Although it could be that they were present on all dates but below the detection limit, their expression was much lower than GH16

enzymes. GH16 and GH17 laminarinases are both endo-acting enzymes, but while GH16 laminarinases have been shown to be highly active on branched laminarin[32,34], GH17 laminarinases show very low activity on the branched molecule and require de-branching by GH30 exo-β-1,6-glucosidases[28,32]. Therefore, a combination of GH17 and GH30 (GH3 aiding total degradation) would deconstruct laminarin into glucose, but GH16 alone can directly hydrolyse the laminarin into oligosaccharides[32]. Overall, the strong and continuous expression of GH16 laminarinases with up to 5.8% NSAF*100 shows that laminarin was actively degraded into oligosaccharides by microbes, which might cause the low persistence of β-1,3-glucan in HMWDOM.

In regard to FCSP, the metagenomic results showed the presence of *Verrucomicrobiae*, which are known degraders of sulphated fucans from brown macroalgae and may also be candidates to degrade the FCSP of diatoms. Degradation of macroalgal FCSP requires sulphatases to remove the extensive sulphate modifications, before GH107 endo-fucanases produce oligosaccharides and exo-fucosidases of the GH29, GH95 and GH141 families hydrolyse these into fucose and other mono-mers[29]. Additionally, carbohydrate esterases and galactosidases have been described as being involved in the degradation of macroalgal FCSP[35]. In the metagenomes we found, at very low abundances, GH107 endo-fucanase genes (Fig. 3b), which encode endo-acting enzymes that can degrade the backbone of sulphated fucans[36–39]. GH107 genes were primarily detected in *Verruco-microbiae*. We also detected genes of exo-fucosidases GH29, GH95 and GH141 (Fig. 3b), most of which were assigned to *Verrucomicrobiae* and *Bacteroidia*. In conclusion, the metagen-ome data showed that *Verrucomicrobiae*, with homologs of enzymes involved in sulphated fucan degradation, were present during the blooms raising the question of whether those were actively degrading the FCSP from diatoms.

The metaproteomic data showed no detection of GH107 endo-fucanases. Of the fucosidase families putatively involved in FCSP degradation, we only detected homologs of bacteroidial GH29 exo-fucosidases in low abundance of 0.08% NSAF*100 at the end of the bloom (Fig. 3b). This suggests there was little to no degradation of FCSP during the blooms. Alternatively *Verrucomicrobiae* abun-dance was, compared with laminarin-degraders, simply too low for their expressed enzymes to be detected as well as to affect FCSP concentration. Sulphated polysaccharides are more stable and less attractive to microbes compared e.g. to laminarin because their degradation takes more effort. FCSP degradation requires a vast collection of sulphatases and other carbohydrate-active enzymes. This high number of different enzymes imposes a high proteome investment on FCSP-degrading bacteria[29].

Supporting evidence for the stability of FCSP stems from previously measured degradation rates, where laminarin and other simple polysaccharides externally added to seawater were hydrolysed faster by bacteria/enzymes than externally-added macroalgal FCSP, which was either not degraded at all or at lower rates in the Arctic, Atlantic and other ocean basins[40]. Taken together, we found that although the microbes encode the putative genetic potential to degrade FCSP (Supplementary Fig. 8), they expressed far less enzymes compared to laminarin-degraders (Fig. 3a, b). These results support the apparent stability of microalgal FCSP over laminarin.

## Discussion
In this study we showed that diatoms produce FCSP that accu-mulates in POM over time, storing carbon during the course of a microalgae bloom period. First the FCSP was coating the cells but was also secreted as soluble polysaccharide and increased in HMWDOM, later it aggregated into particles and increasingly

coated the surface of diatoms in a mucin-like manner. The solubility of the secreted FCSP could be concentration dependent, since assembly of unknown anionic dissolved polysaccharides into exopolymer particles has been previously described in terms of polymer gel theory and aggregation models, which are con-centration dependent[7,8]. Alternatively, it might be that FCSP was modified biochemically, increasing adhesiveness and aggregation over time. Here, by adhesive polysaccharides we refer to glycans that have physicochemical features, such as sulphation, that enhance their assembly properties and may lead to formation of TEP-like particles.

The biological role of diatom FCSP remains unclear. When nutrient and light conditions are favourable diatoms can bloom and since light is so important for photosynthesis, they regulate their buoyancy to remain suspended in the surface ocean where they can grow[10]. However, aggregation of diatoms into particles would make them sink. There are two major theories for the formation of marine gel particles. According to the polymer gel theory, polymers assemble into polymer networks when the dis-tance between them allows interaction by chemical or physical cross-links, therefore the concentration and the physicochemical characteristics of the polymers are key for the process[41]. For instance, dissolved anionic polysaccharides can spontaneously assemble into polymer gels (such as TEP) by cross-links with cations like calcium[7]. According to the coagulation theory, mul-tiple factors, such as particle concentration, stickiness, differential sedimentation and collision rate, contribute to the formation of aggregates (particles first collide and then stick to each other) that drive the biological carbon pump[42,43]. Both theories have been supported empirically[41] and the corresponding processes could be complementary. One of the factors that promote aggregation is TEP, which is highly sticky and has been proposed to act as a glue between diatom cells[8,44]. Therefore, (i) formation of TEP-like[45] particles would most likely promote aggregation of diatoms and (ii) having an adhesive polysaccharide (FCSP) surrounding the diatom cell would possibly enhance this aggregation, since cell stickiness would make each collision more efficient. This increased aggregation contributes to their vertical transport, because aggregated cells sink faster than single cells. Additionally, the chemical groups that drive polysaccharide aggregation via intermolecular ionic bonds with cations, also contribute to trace metal cycling. Anionic algal polysaccharides have high binding affinities to metals[46], therefore aggregation of these poly-saccharides as well as incorporation of trace metals into sinking particles result in removal of biogeochemical relevant trace metals from surface waters[47].

In regard to diatom exudates, it has been shown that the concentration of dissolved adhesive exudates that generate TEP coincides with flocculation of diatoms towards the end of blooms[6,44]. Although aggregation and sinking at the end of blooms is part of the lifecycle of most planktonic diatoms, we observed high secretion of dissolved FCSP since the beginning of the diatom bloom, when the photosynthetic cells need to be in the euphotic zone in order to be able to grow and bloom. In this context, the synthesis and secretion of adhesive polysaccharides already at early bloom phases appears detrimental to diatoms and raises questions over their biological function. While it is well-known that diatoms and other microalgae release dissolved organic molecules (the largest fraction being polysaccharides[48]) into the surrounding seawater, the reasons behind it are not fully understood. Different mechanisms, which are not mutually exclusive, have been proposed to explain microalgae exudation, including passive loss where the release of small uncharged molecules is driven by passive diffusion through the cell mem-brane[49], as well as mechanisms with active transport of the molecules to the environment. It is suggested that a major reason

for the active release of polysaccharides by microalgae cells is an overflow mechanism to funnel reducing equivalents from photosynthesis into photosynthate, i.e. active removal of excess fixed carbon when photosynthesis exceeds the requirements for growth[50]. However, the structure of FCSP with its extensive sulphation and various sugar monomers appears quite complicated for a molecule that simply serves to funnel away reduced carbon.

The observed stability against bacterial degradation, and the structure and physicochemical behaviour of diatom FCSP point towards specific biological functions. Many species of brown macroalgae, which are evolutionarily related to diatoms, contain structurally related types of FCSP that are part of their cell walls[51] and also part of their released exudates[52]. Likewise, we found that diatoms secrete FCSP, a fraction of which remains outside the cell, coating its surface, and a fraction is released into the HWMDOM pool. Except for some marine animals (echinoderms), the presence of FCSP seems so far restricted to brown macroalgae and diatoms. Brown macroalgal FCSPs have been shown to have antiviral and antibacterial activity[53], suggesting they might play a role in viral defence and in modulating the algal microbiome close to or on the macroalgal surface phycosphere. By analogy, plant roots secrete structurally complex mucilage polysaccharides into the surrounding rhizosphere which preferentially promote growth of beneficial bacteria[54]. Plant cells secrete anionic pectic galacturonans forming a gel barrier prohibiting bacterial digestive enzymes from reaching and degrading the cell wall[55]. In animals, epithelial cells of the intestine secrete mucins, highly glycosylated proteins, forming a barrier against invasion by commensals and pathogens[56]. Common to these glycopolymers is their high structural complexity and the presence of anionic carboxyl or sulphate groups. Given its stability against degradation, FCSP, which coats the diatom cells, may function as a barrier protecting the cell wall against microbes and their digestive enzymes.

While a defensive barrier of FCSP might have a positive effect on population size, the increasing concentration of secreted adhesive polysaccharides leads to aggregation of diatoms into particles, which in turn promotes their sinking. Thus, self-aggregation of diatoms would reduce population size. The more adhesive the polysaccharide, the better it forms a gel-like barrier around diatom cells or encapsulates pathogens. Yet, (in addition to the aggregation driven by FCSP-containing particles) increased stickiness would also increasingly draw down diatom cells through self-aggregation. Therefore, the secretion of FCSP since early bloom phases resembles a tradeoff. Although diatom secretion of FCSP might be driven by constitutive production with no control by microbes, interactions between diatoms and microbes can modulate the diatom exudates. For instance, some strains of attached bacteria have been found to induce diatom production of exudates and TEP formation[57]. Bacteria have also been shown to affect the composition of diatom exudates[58]. Microbial interactions might therefore shape the evolution of this potential defence-aggregation tradeoff and thereby regulate carbon export via the biological carbon pump. The presence of a stable polysaccharide that coats labile molecules within algal cells or particles provides an unexplored mechanism for the stabilisation of organic matter in the ocean.

## Methods

**Field sampling for POM and HMWDOM.** Sampling was carried out from the 15th of March to the 26th of May in 2016 during a phytoplankton bloom period at station Kabeltonne in the North Sea (54°11.3′N, 7°54.0′E) near the island of Helgoland, Germany. Two times per week (Tuesdays and Thursdays) 100 L of subsurface seawater (1 m depth) were collected in the morning with the research vessel Aade. See sampling scheme in Fig. 1b. For harvesting POM samples, the 100 L seawater were directly transported to the lab in 25 L carboys and sequentially

filtered through 10 μm, 3 μm and 0.2 μm pore size polycarbonate filters with 142 mm diameter (filter codes were TCTP14250, TSTP14250 and GTTP14250, respectively; Merck Millipore). To reduce the filtration time, filtration was performed sequentially but in parallel using three filtration units (Pressure filter holder type 16275, Sartorius) with air pressure pumps (Flojet G57, ITT Industries) by <2 bar. The entire filtration was normally completed in <4 h after sampling. Filters were replaced for new filters when clogged and all were stored at −80 °C until further analysis. For harvesting HMWDOM samples, 100 L of 0.2-μm-filtered seawater (obtained after the sequential filtration described above) were concentrated to a final volume of 0.5 L by tangential flow filtration (TFF). The TFF system (Sartocon slice stainless steel filter holder, Sartorius Stedim) was run with 3 filter cassettes with a cut-off of 1 kDa (3051460901E–SG, Sartocon® Slice PESU Cassettes, Sartorius Stedim) and an air pressure pump (Flojet G57, ITT Industries) using feed and retentate pressures of <4 bar and with the permeate valve opened. The obtained concentrated samples correspond to high molecular weight dissolved organic matter, HMWDOM. Once the retentate had a volume of ~0.5 L, the TFF was stopped and the retentate sample (containing molecules <0.2 μm and >1 kDa) was collected in a 0.5-L Nalgene bottle. The sample was transferred to 1 kDa dialysis tubing (Spectra/Por®, Spectrum Laboratories) and dialysed at 4 °C overnight against 10 L deionised water with stirring (sample was further dialysed after the field campaign, see Processing of HMWDOM samples section), collected again in a 0.5-L Nalgene bottle and stored at −80 °C until further analysis. In between runs the TFF system and filter cassettes were washed according to the manufacturer's recommendations with 4 L of 1 M NaOH at 50 °C for 1 h; afterwards the system was flushed with 0.2-μm-filtered tap water and the filter cassettes permeated with deionised water until pH of permeate was neutral. Sampling dates were the following: 2016/03/15, 2016/03/17, 2016/03/22, 2016/03/24, 2016/03/29, 2016/03/31, 2016/04/05, 2016/04/07, 2016/04/12, 2016/04/14, 2016/04/19, 2016/04/21, 2016/04/26, 2016/04/28, 2016/05/03, 2016/05/10, 2016/05/12, 2016/05/17, 2016/05/19, 2016/05/24 and 2016/05/26.

**Dissolved organic carbon analysis.** Samples for quantification of DOC were collected from the 0.2-μm-filtered seawater. Each sampling day, four replicate samples of 10 mL were acidified to pH 2 (HCl, p.a.) in pre-combusted glass vials (400 °C, 4 h) and stored in the dark at 4 °C until analysis. DOC was analysed on a Shimadzu TOC-VCPH instrument via high temperature catalytic oxidation. Analytical accuracy and precision were tested with deep-sea reference material (D. Hansell, University of Miami, USA) and were better than 5%. Standard deviation for replicates was on average 2.8 ± 1.6%. DOC analysis of the 0.5 L HMWDOM samples was also performed. Based on the DOC concentrations of the DOM (0.2-μm-filtered seawater) and HMWDOM samples, we determined that by TFF we recovered on average 5.5% of the DOM.

**Quantification of monosaccharides with HPAEC-PAD.** We adapted a protocol for determination of neutral, amino and acidic sugars using high performance anion exchange chromatography (HPAEC) described previously[15]. In short, a Dionex ICS-5000+ system with pulsed amperometric detection (PAD) was equipped with a CarboPac PA10 analytical column (2 × 250 mm) and a CarboPac PA10 guard column (2 × 50 mm). System control and data acquisition were performed with Chromeleon v7.2 software (Dionex). Neutral and amino monosaccharides were separated by an isocratic flow of 18 mM NaOH for 20 min, followed by a gradient of 200 mM sodium acetate to separate acidic sugars. Monosaccharide standards were used to identify peaks by retention time and a standard mix ranging from 1–10 to 1000 μg L$^{-1}$ was used to quantify the amount of monosaccharide ($x$ axis amount and $y$ axis peak area).

*Monosaccharide analysis of HMWDOM.* From each of the 0.5 L TFF-concentrated HMWDOM (except for dates 2016/03/15 and 2016/05/03, total of 19 samples), 150 μL of sample were acid hydrolysed at 100 °C for 24 h with 150 μL of 2 M HCl (7647-01-0, Analar Normapur) in pre-combusted glass vials (400 °C, 4 h). Afterwards, two 100 μL aliquots of each acid hydrolysis were taken for technical replication and dried for 1.5 h at 30 °C in a centrifugal vacuum concentrator. For this analysis, 100 μL aliquots of monosaccharide standard mix were prepared in 1 M HCl and were dried together with the samples in the centrifugal vacuum concentrator. Samples and standards were resuspended in 100 μL MilliQ for subsequent HPAEC-PAD analysis. Standard deviation for replicates was on average 8%.

*Monosaccharide analysis of purified FCSP.* The 1 mL anion exchange chromatographic fractions with purified FCSP (fractions with highest mAb BAM1 absorbance peak - for the bloom samples the dates were 2016/03/17 for HMWDOM and 2016/05/12 for POM) were collected and in order to get rid of the salt they were dialysed in 1 kDa dialysis tubing (Spectra/Por®, Spectrum Laboratories) at 4 °C against 500 mL MilliQ in beakers with stirring, replacing the water (at least 4 h before every water replacement) three times. Once desalted, 400 μL of the sample were acid hydrolysed with 400 μL of 2 M HCl. Two independent acid hydrolysis were performed per each chromatographic fraction. Samples were prepared and analysed as described above, except for that samples were not 2-fold diluted when

resuspended (300 μL aliquots of each acid hydrolysis were dried and resuspended in 150 μL MilliQ). Standard deviation for replicates was on average 7.2%.

**Quantification of sulphate**. The sulphate released by acid hydrolysis of the purified FCSP (same chromatographic fractions and same sample preparation as described above in the Monosaccharide analysis of purified FCSP section) was measured on a Metrohm 761 Compact ion chromatograph equipped with a Metrosep A Supp 5 analytic column and suppressed conductivity detection with 0.5 M $H_2SO_4$. System control and data acquisition were performed with MagIC Net v3.2 software (Metrohm). Ions were separated by an isocratic flow of carbonate buffer (3.2 mM $Na_2CO_3$ and 1 mM $NaHCO_3$) with sulphate eluting at 16 min and a duration of 20 min per run. Standard deviation for replicates was on average 4.7%.

**Processing of POM samples**. For POM samples, one individual filter (of 10, 3 or 0.2 μm pore size) was placed in a clean Polystyrene Petri dish with 145 mm diameter and the material retained on it was scraped out from the filter with a flat spatula, washed out from the filter surface with pure ethanol and collected in a 50 mL Falcon tube. To ensure the removal of the filters content, this procedure was repeated three times per each single filter. Per each sample (one specific pore size and date), material was extracted from three replicate filters and pooled together (therefore a total of 189 filters were processed). The samples material corresponds to POM with the equivalent molecular sizes of over 10 μm, between 10 and 3 μm, and between 3 and 0.2 μm. The alcohol insoluble residue (AIR) was performed for all POM samples as follows: the scraped out, ethanol washed material that was collected in 50 mL Falcon tubes was vortexed and then rotated at room temperature for 10 min. Afterwards samples were spun down at $3480 \times g$ for 15 min at 15 °C. Supernatants were discarded and pellets were resuspended (6:1 volume solvent:volume pellet) in chloroform:methanol (1:1). Samples were vortexed and rotated for 10 min and then spun down as described above. Supernatants were discarded and pellets were resuspended in pure acetone (6:1 volume solvent:volume pellet), vortexed and rotated for 10 min and spun down as described above. Supernatants were discarded and pellets were left to air dry under a fume hood overnight.

**Processing of HMWDOM samples**. The TFF-concentrated HMWDOM samples were further concentrated from 0.5 L to ~40 mL by using an Amicon® stirred ultrafiltration cell (Merck Millipore) with a 1-kDa membrane (76 mm PES, Sartorius) on ice using nitrogen for gas pressure. In order to get rid of the salt, the concentrated sample was transferred to a 1-kDa dialysis tubing (Spectra/Por®, Spectrum Laboratories) and dialysed at 4 °C overnight against 5 L MilliQ in a beaker with stirring. Subsequently, the MilliQ was replaced twice with new 5 L MilliQ with at least 4 h stirring in between replacements. After the third replacement, the desalted HMWDOM sample was frozen and freeze dried.

**Polysaccharide extraction**. As polysaccharides have different extractability and solubility we extracted them with three different solvents. All the HMWDOM freeze dried samples and the POM AIR samples were individually well-homogenised with a spatula. Samples were weighed out in 8-strip tubes (approximately 10 mg of each sample) and a 1.6-mm stainless steel bead was added in each tube to aid sample mixing. Polysaccharides were sequentially extracted with: autoclaved MilliQ water, 50 mM EDTA pH 7.5 and 4 M NaOH with 0.1% w/v NaBH₄. For each of the extracting solvents the following was performed: for every 10 mg of sample 300 μL of solvent were added to the tube (volume of solvent was adjusted to the weight i.e. sample extractions were normalised by weight). The tubes - set in an appropriate holder - were placed on a TissueLyser II (Qiagen) with which samples were shaken at 30/s for 2 min and then at 6/s for 2 h. After the 2 h extraction, samples were spun down at $6000 \times g$ for 15 min at 15 °C. Extracts (supernatants) were collected in 1.5 mL tubes and stored at 4 °C. The pellets were resuspended in the next extracting solvent using the same extraction procedure (with the TissueLyser II) as depicted above. The three sequential extractions of all the samples were performed on the same day.

**Carbohydrate microarray analysis**

*Microarray printing*. All HMWDOM and POM polysaccharide extracts were added into wells of 384-microwell plates. For each extract a 2-fold dilution followed by a 5-fold dilution was performed in printing buffer (55.2% glycerol, 44% water, 0.8% Triton X-100). Plates containing the samples were spun down at $3500 \times g$ for 10 min at 15 °C to get rid of bubbles. The content of the plates was printed onto nitrocellulose membrane with a pore size of 0.45 μm (Whatman) using a micro-array robot (Sprint, Arrayjet, Roslin, UK) under controlled conditions of 20 °C and 50% humidity. A printing replicate was included for each sample and therefore each extract was represented by a total of 4 spots in the microarray (two dilutions per extract and a replicate for each). Note: the HMWDOM water extracts presented very high viscosity, probably due to a high polysaccharide content, and solutions with very high viscosity have a negative effect on the quality of the microarray print (appearance of satellites). To guarantee a good quality print, the HMWDOM water extracts were 3-fold diluted in MilliQ water prior to being 2-fold diluted in printing buffer. Numerous identical microarrays (populated with the same time-series extracts) were printed.

*Microarray probing*. The printed microarrays were blocked for 1 h in phosphate buffered saline (PBS, 1x) with 5% (w/v) non-fat milk powder (MPBS). Then, MPBS was discarded and each single microarray was individually incubated for 2 h with one of 51 polysaccharide-specific monoclonal antibodies (mAbs) or carbohydrate binding modules (CBMs) listed in Supplementary Table 1 (see mAb suppliers in Reporting Summary; CBM3a and CBM4-1, PlantProbes; CBM27 and CBM30, NZYTech). The mAbs from PlantProbes, CarboSource and INRA were diluted 1:10; mAbs from BioSupplies 1:1000; CBMs 10 μg mL⁻¹ in MPBS. Exceptions: PAM1 10 μg mL⁻¹, 2F4 1:250 and INRA-AX1 1:25 in MPBS. After the 2 h incubation, arrays were thoroughly washed in PBS and incubated for 2 h with either anti-rat, anti-mouse or anti-His tag secondary antibodies conjugated to alkaline phosphatase diluted 1:5000, 1:5000 or 1:1500, respectively, in MPBS. After thorough washing in PBS, arrays were washed in deionised water and then developed in a solution containing 5-bromo-4-chloro-3-indolylphosphate and nitro blue tetra-zolium in alkaline phosphatase buffer (100 mM NaCl, 5 mM MgCl₂, 100 mM Tris-HCl, pH 9.5).

*Microarray analysis*. The developed arrays were scanned at 2400 dots per inch and binding of each probe (probe signal intensity) against each spotted sample was quantified using the software Array-Pro Analyzer 6.3 (Media Cybernetics). For each extract the mean antibody signal intensity (derived from 4 spots) was calculated. The highest mean signal intensity detected in the data set was set to 100 and all other values were normalised accordingly. Note that HMWDOM and POM data sets required independent normalisation as they required different sampling strategies, therefore the temporal dynamics of polysaccharides can be compared between the two pools but not the absolute number. In contrast, absolute numbers of epitope abundance detected by one particular probe within HMWDOM samples (water extracts independently from EDTA and NaOH, due to the waters additional dilution) or within all POM samples can be compared. Quality of printed micro-arrays was verified; spots had a uniform size, of correct morphology and alignment. Controls for the extraction solvents indicated no unspecific binding for any of the probes and controls for alkaline phosphatase-conjugated secondary antibodies (anti-rat, anti-mouse and anti-His tag) presented no unspecific binding to any of the samples. A cut-off of 5 arbitrary units was applied and Supplementary Fig. 5 shows all profiles where in at least one date an antibody positive signal (value ≥5) was detected, error bars denote the standard deviation of printing replicates.

**Antibodies and CBMs**. All primary antibodies and CBMs used in this study are listed in Supplementary Table 1. They are commercially available at: PlantProbes, Leeds, UK; BioSupplies, Bundoora, Australia; CarboSource, Athens, USA; NZY-Tech, Lisbon, Portugal. Except for the INRA antibodies that were provided by the Institut National de la Recherche Agronomique, Nantes, France. References of publications that validate their specificities are provided in Supplementary Table 1. The secondary antibodies used for the different experiments of this study include: anti-rat, anti-mouse and anti-His tag secondary antibodies conjugated to alkaline phosphatase (A8438, A3562 and A5588, respectively, Sigma-Aldrich); anti-rat conjugated to FITC (F1763, Sigma-Aldrich); anti-rat conjugated to peroxidase (A9037, Sigma-Aldrich).

**Immunofluorescence microscopy**

*Immunolabelling*. Immunolabelling of filters from the beginning and end of the bloom for microscopy analyses was performed four times independently, including filters from the dates 2016/03/24, 2016/03/29, 2016/03/31, 2016/05/10, 2016/05/12 and 2016/05/19. For sample collection, subsurface seawater was sampled (in parallel with the 100 L to get the same water body) and 1 L was fixed, without pre-filtration, with 37% formaldehyde to a final concentration of 1% (v/v) for 1 h at room temperature. Once fixed, triplicates of 100 mL were filtered onto poly-carbonate filters with 0.2 μm pore size and 47 mm diameter (GTTP, Merck Millipore) with a manual vacuum pump with <5 mm Hg. The filters, which contained all POM fraction >0.2 μm, were stored at −80 °C until analysis. For immunola-belling, sections of the filters were cut and added into separate 2 mL Eppendorf tubes. Filter sections were blocked in PBS with 3% (w/v) bovine serum albumin (BSA-PBS) for 1 h. Next, sections were incubated for 1.5 h with the rat mAb BAM1 (PlantProbes) diluted 1:5 in BSA-PBS. Then they were washed four times with PBS and incubated with anti-rat secondary antibody conjugated to FITC (F1763, Sigma Aldrich) diluted 1:100 in BSA-PBS for 1.5 h in darkness. The thorough washing step with PBS was repeated. Afterwards, filter sections were incubated with 4′,6-diamidino-2-phenylindole (DAPI, at 1 μg mL⁻¹) for 10 min in the dark, flushed with MilliQ and then in 96% ethanol. Filter sections were mounted on glass slides with the antifading reagents Citifluor:Vectashield as 4:1 (v/v). Control filters with no BAM1 incubation as well as controls with no BAM1 neither secondary anti-rat-FITC nor DAPI were included; to verify absence of unspecific binding of the secondary antibody and to test sample auto fluorescence, respectively.

*Super-resolution microscopy by Airyscan*. The immunolabelled filters were examined with a confocal laser scanning microscope LSM 780 (Zeiss) equipped with Airyscan detector (Carl Zeiss, Germany). DAPI, FITC and diatom auto fluorescence were excited using laser lines of 405, 488 and 561 nm wavelength, respectively. The detection windows for DAPI, FITC and diatom auto fluorescence

emission were set to 420–460 nm, 495–550 nm and long pass from 605 nm, respectively. Images were collected and processed using the ZEN black software (Zeiss). Z-stack images were taken with a 63x/1.4 oil objective. Movement of the sample during acquisition was corrected by Z-stack alignment (with the options: 2Dslices, highest resolution, translation and linear) with the ZEN blue software (Zeiss). Images were projected as maximum intensity projection in ZEN black software and representative images from dates 2016/03/31 and 2016/05/10 are shown in Fig. 2a, b.

*Quantification of FCSP intensity.* Immunolabelled filters were examined on a Zeiss Axio Imager D2 epifluorescence microscope equipped with an AxioCam MRm camera (Carl Zeiss, Germany). Images were taken on the same day using Zeiss AxioVision software with a 63x/1.4 oil objective and keeping the excitation lamp intensity stable and same exposure time for all the filters. BAM1 signal intensity was measured through the secondary anti-rat FITC-conjugated antibody, using filter set F36-525 with excitation at 472/30 nm, beam splitter 495 nm and emission window at 520/35 nm (AHF-Analysentechnik, Germany). Signal intensity was quantified both on diatom cells and on areas of the filter with no diatoms. The Intensity Mean Value of these regions of interest, which were manually defined, was quantified using the ZEN blue software. The dates 2016/03/31 and 2016/05/10 were chosen (highest chlorophyll a value from sampling dates of March and May, 9.0 and 7.0 mg m$^{-3}$, respectively) to do the comparison in FCSP signal between the beginning and end of the bloom.

**Light micrographs.** To have an equivalent bright-field image of diatom cells from the same species (Inset in Fig. 2a), aliquots of an 8-day-old laboratory culture of *Chaetoceros socialis* (same growing conditions as described below in the Diatom laboratory cultures section) were examined. Samples were not fixed neither filtered and 75 μL of the diatom culture were directly mounted on glass slides an examined on a Nikon 50i microscope equipped with an AxioCam MRc camera (Carl Zeiss, Germany). Images were taken with a 60x/1.4 Plan Apo oil objective, operating with bright-field optics and using Zeiss AxioVision software.

**Total microbial cell counts.** Total microbial cell numbers were assessed on a weekday basis and were carried out as described previously[14]. Briefly, subsurface seawater was fixed with 1% (v/v) formaldehyde and filtered through 0.2 μm polycarbonate filters (47 mm diameter, GTTP, Merck Millipore). Filter sections were stained with DAPI (at 1 μg mL$^{-1}$) and labelled microbial cells were counted using an automated epifluorescence microscope via the automated image acquisition and analysis system described previously[59].

**Diatom laboratory cultures.** The centric diatom *Chaetoceros socialis* was isolated at station Kabeltonne during the 2016 spring bloom by Tilmann Harder group (University of Bremen, Germany) at the same time that we were harvesting our POM and HMWDOM samples. Monospecific cultures of this diatom strain as well as of *Thalassiosira weissflogii* and *Nitzschia frustulum* (Bigelow Laboratory) were grown in ESAW media[60] in an incubation chamber at 15 °C on a constant 12 h:12 h light:dark cycle and without shaking. Growth light intensity was applied with five Sylvania F36W/GRO tubes, resulting in an irradiance of ~140 μmol photons m$^{-2}$ s$^{-1}$. Diatom cultures were prepared by inoculating 7 mL of a 10-day-old culture to 1 L ESAW media. Per each of the three diatom species, four 1 L non-axenic cultures were harvested at 10 days after inoculation (at stationary phase) by centrifugation at 6800 × *g* for 20 min at 15 °C. The cell pellets, diatom biomass, were collected and freeze dried. Samples were homogenised with a spatula and AIR was performed as described above.

**FCSP separation by anion exchange chromatography.** Polysaccharide extraction from HMWDOM, POM (3 μm filter) and diatom laboratory cultures (*C. socialis, T. weissflogii* and *N. frustulum*) biomass was performed as described above but with a ratio mg biomass:mL solvent of 2:1, 4:1 and 10:1, respectively. FCSP from crude MilliQ water extracts was separated by anion exchange chromatography by injecting 1 mL aliquots of the extracts (HMWDOM extracts required a 5-fold dilution in 20 mM Tris buffer to avoid column overload) to a 1 mL Hi-Trap ANX FF column (GE Healthcare) using a Bio-Rad chromatography system. Polysaccharide separation by chromatography and detection using enzyme-linked immunosorbent assay (ELISA) analysis was performed as described previously[61]; but using 20 mM Tris pH 7.5 together with 20 mM Tris 5 M NaCl pH 7.5 as buffers (50 mM sodium acetate pH 5.0 with 50 mM sodium acetate 5 M NaCl pH 5.0 buffers were proven successful for FCSP separation as well) and using the rat mAb BAM1 as detection tool to identify chromatographic fractions containing FCSP. Absorbance at 450 nm (mAb BAM1 signal) was determined using a Spectro-starNano absorbance plate reader and MARS software (BMGlabtech) obtaining the values as optical density. Figure 2e shows single representative chromatography runs from the dates 2016/03/31 and 2016/05/12 for both pools. Chromatography plus ELISA analyses of extracts from each of the two dates was independently performed four and two times for HMWDOM and POM, respectively, showing high reproducibility. Developing time for ELISA differed for different runs as this method was used to detect the AEC fraction containing the purified FCSP and not to compare its concentration among different extracts.

## Metagenomics

*Metagenomic analysis.* Samples for the dates 2016/03/16, 2016/03/21, 2016/03/31, 2016/04/12, 2016/04/19, 2016/04/26, 2016/05/02, 2016/05/12 and 2016/05/17 were collected and sequenced as described previously[62]. Briefly, seawater was pre-filtered sequentially to remove eukaryotes using both 10 μm and 3 μm pore size filters, and then bacterioplankton was collected on 0.2 μm pore size polycarbonate filters (0.2–3 μm size fraction). DNA was sequenced at the DOE Joint Genome Institute (JGI) on the Illumina HiSeq 2500 platform with 2 × 150 bp reads. Raw reads were quality filtered and trimmed using BBDuk v35.14 (http://bbtools.jgi.doe.gov) and the following parameters: ktrim = r k = 28 mink = 12 hdist = 1 tbo = t tpe = t qtrim = rl trimq = 20 minlength = 100. Reads were assembled with metaSPAdes v3.10.0, with kmer lengths of 21, 33, 55, 77 and 99. Contigs longer than 2500 bp were kept for further analyses. Assemblies are available under the accession PRJEB28156. Gene prediction and translation was done with Prodigal v2.6.3[63] in metagenomic mode. Amino acid sequences were then initially placed into CAZyme families using the hmmscan function of HMMer v3.2.1[64] and the dbCAN v6 database and using the hmmscan-parser.sh script to select the best hit[65]. Predicted CAZymes were then confirmed via DIAMOND-BLAST comparison[66] selecting the top hit against the CAZy database v07312018[67], using –sensitive mode, an e-value cut-off of 1$^{-20}$, minimum query cover of 40%, and minimum sequence identity of 30%[33]. In order to estimate abundance of different gene families, reads were mapped to the individual CAZyme gene sequences using BBMap v35.14 (http://bbtools.jgi.doe.gov), with minimum mapping identity of 0.99, and identity filter for reporting mappings of 0.97. Reads per kilobase per million (RPKM = 1,000,000 × (number of reads mapped/gene length in kilobase pairs)/number of reads in sample) values were calculated to estimate the normalised relative abundance of individual gene families.

*Classification of MAGs.* CAZymes were assigned to taxonomic groups based on their presence in MAGs from PRJEB28156. MAGs were classified using GTDB-Tk v0.3.1 and GTDB v89[68]. Classifications are those of the GTDB[68], and therefore the class *Flavobacteriia* is subsumed along with *Cytophagia* by the class *Bacteroidia*. *Bacteroidia* is one of two classes in the GTDB phylum *Bacteroidota* (elsewhere known as *Bacteroidetes*), the other class being *Rhodothermia*. The class *Verruco-microbiae* excludes the class *Lentisphaeria*, both of which are part of the GTDB phylum *Verrucomicrobiota*.

## Metaproteomics

*Sample preparation.* Samples were collected on 2016/03/17, 2016/03/31, 2016/04/19, 2016/05/03, 2016/05/12 and 2016/05/17 by the same sampling procedure as described above in the Metagenomic analysis section. Proteins from the 0.2 μm pore size polycarbonate filters were extracted using a modified protocol[69]. Briefly, one eighth of a filter (142 mm diameter) was cut into small pieces and transferred into tubes with 100 μL resuspension buffer 1 (50 mM Tris-HCl pH 7.5, 0.1 mg mL$^{-1}$ chloramphenicol, 1 mM phenylmethylsulfonyl fluoride (PMSF)). Next, these samples were incubated in 150 μL resuspension buffer 2 (20 mM Tris-HCl pH 7.5, 2% SDS) for 10 min at 60 °C and 1200 rpm. Samples were mixed with 500 μL DNAse buffer (20 mM Tris-HCl pH 7.5, 0.1 mg mL$^{-1}$ MgCl$_2$, 1 mM PMSF, 1 μg mL$^{-1}$ DNAse I) and cells were subsequently disrupted by ultra-sonication (4 × 2 min, amplitude 51–60%, cycle 0.5). The lysate was incubated for 10 min at 37 °C and 1200 rpm. Samples were spun down twice at 10,000 × *g* for 10 min at 4 °C and protein extract (supernatant) from both centrifugation steps were collected in the same tube. Proteins were precipitated by addition of pre-cooled trichloroacetic acid (TCA, 20% v/v). After centrifugation (12,000 × *g*, 30 min, 4 °C), protein pellets were washed in pre-cooled (−20 °C) acetone (3 × 10 min, 12,000 × *g*, 4 °C) and vacuum-dried. Proteins were resuspended in 2x SDS sample buffer (4% SDS (w/v), 20% glycerine (w/v), 100 mM Tris-HCl pH 6.8, bromphenol blue, 3.6% 2-mercaptoethanol (v/v)), separated by 1D SDS-PAGE and stained with Coomassie Brilliant Blue. Gel lanes were cut in 20 equal pieces (for the first 2 replicates) or in 10 pieces (for the third replicate), de-stained, and trypsin-digested as described previously[14].

*LC-MS/MS analysis.* Liquid chromatography-tandem mass spectrometry (LC-MS/MS) measurements were performed using an Easy-nLC 1200 coupled to an Orbitrap Elite mass spectrometer (both Thermo Fisher Scientific). Peptides were loaded onto in-house packed capillary columns filled with 3 μm ReproSil-Pur 120 C18-AQ (Dr. Maisch GmbH, Germany) and separated using a 156-min (for the first two replicates) or a 226-min (for the third replicate) nonlinear binary gradient from 1% to 99% solvent B (95% (v/v) acetonitrile, 0.1% (v/v) acetic acid) in solvent A (0.1% (v/v) acetic acid) at a constant flow rate of 300 nL min$^{-1}$ and 45 °C. A survey scan (resolution 60k at *m/z* 400) was performed, followed by CID fragmentation of the 20 most abundant precursor ions (top20). Dynamic exclusion was enabled. Resulting MS/MS spectra were searched against a target-decoy database containing all protein sequences from the nine metagenomes obtained during the studied spring bloom 2016 (from which redundant sequences were removed using cd-hit at 99% identity[70] and a set of common laboratory contaminants was added) using Mascot (version 2.6.0). All protein sequences in the database were reversed and appended to the database to allow for false discovery rate (FDR) calculation. Peptide identifications (greater than 95.0% probability, Scaffold Local FDR algorithm) and protein identifications (greater than 99.0% probability, at least 2

exclusive unique peptides) were merged and validated with Scaffold (version 4.8.6). Additionally, an X! Tandem search was performed (The GPM, thegpm.org; version CYCLONE (2010.12.01.1)) with default settings. Protein probabilities were assigned by the Protein Prophet algorithm[71]. Percent normalised spectral abundance factor (%NSAF) values were calculated based on the number of total spectral counts obtained per protein group in each sample[72]. Average values from the three biological replicates were calculated (proteins that were not identified in one of the replicates were included as "0" value in the calculation).

*CAZyme annotation.* CAZyme family assignments used for detected proteins were those generated in the above Metagenomic analysis section. CAZymes were assigned to taxonomic groups as described above in the Classification of MAGs section.

**Physicochemical parameters and phytoplankton measurements**. Data were assessed from subsurface seawater at station Kabeltonne as part of the Helgoland Roads LTER time series[73], which can be accessed via the open database PANGAEA (https://www.pangaea.de). All physicochemical parameters as well as phytoplankton taxa and species measurements were done on a weekday basis. Chlorophyll a concentration was measured using an algal group analyser (bbe moldaenke). Details on data acquisition have been described previously[14]. For phytoplankton cell counts: subsurface seawater samples were preserved in 0.1% neutral Lugol's iodine in amber glass bottles. Depending on the cell density 25 or 50 mL samples were counted using the Utermöhl method[74] with an inverted microscope AxioObserver A1 (Zeiss). Phytoplankton (including nano-phytoplankton and micro-phytoplankton) were counted and identified to species or genera level when possible, and otherwise at taxa level. Samples were examined under 50x to 400x magnification depending on the cells size. Depending on the cell abundances the whole area of the counting chamber or 1 to 4 fields were counted.

**Microarray substrate concentration effect experiment**. To show the correlation between epitope concentration and probe signal intensity the following commercial defined polysaccharides were used: fucoidan (*Laminaria*, PSa13) from Glycomix; galactomannan (carob), glucomannan (konjac) and polygalacturonic acid (PGA, citrus pectin) from Megazyme. The defined polysaccharides were dissolved in MilliQ water to 2 mg mL$^{-1}$. The polysaccharide solutions were added into wells of 384-microwell plates including eight successive 2-fold dilutions in printing buffer. Each polysaccharide solution had twelve replicates for each of the eight different concentrations to also assess spot reproducibility. Microarray printing, probing and analysis was performed as described in Carbohydrate microarray analysis section.

**Quantitative ELISA**. To determine the FCSP concentration fold change, quantitative ELISA was performed with a macroalgae fucan as standard. HMWDOM freeze dried samples and POM (3 μm filter) AIR samples were weighed out in 2 mL tubes and a 1.6-mm stainless steel bead was added to aid sample mixing. Polysaccharide extraction was performed with a ratio of 2 mg biomass:1 mL MilliQ water in a heating block at 25 °C at 650 rpm for 2 h. Samples were spun down at 6000 × g for 15 min at 15 °C. Extracts (supernatants) were analysed by ELISA including a series of 2-fold dilutions in PBS to avoid sample saturation. HMWDOM and POM extracts had five and three 2-fold dilutions in PBS, respectively. The reference standard was fucoidan (*Laminaria*, PSa13, Glycomix) dissolved in MilliQ water and a serial dilution of known concentrations was prepared in PBS. Extracts and standards were added in triplicates at 100 μL per well in 96-well plates (NUNC) and incubated at 4 °C overnight. ELISA analysis was performed as described previously[61] using rat mAb BAM1. A fucoidan standard curve for HMWDOM and one for POM, with $R^2$ values of 0.98 and 0.99 respectively, were used to determine the concentrations of the extracts as ng macroalgal fucan equivalent per mL extract.

**Statistical analyses**. Immunolabelling experiments were performed four times including replicates and filters from six different dates (dates stated in Immunolabelling section), which resulted in comparable observations between experiments. To test for significant changes in FCSP abundance between beginning and end of the bloom, both on diatom cells and on particles, we used two-sided Welch's *t*-test, which was performed using SciPy v1.4.1 (scipy.stats.ttest_ind) and Python v3.7.

**Reporting summary**. Further information on research design is available in the Nature Research Reporting Summary linked to this article.

## Data availability
Glycan analyses data are available in PANGAEA, the carbohydrate microarray data are available at https://doi.pangaea.de/10.1594/PANGAEA.924287 and the monosaccharide composition data are available at https://doi.pangaea.de/10.1594/PANGAEA.924264. Metagenome assemblies and metagenome assembled genomes are available in the European Nucleotide Archive project PRJEB28156. Proteome mass spectral data are available in the ProteomeXchange Consortium via the PRIDE partner repository with the identifier PXD019294. All other data are available within the article and supplementary

information files or from the corresponding author on reasonable request. The publicly available databases dbCAN v6 [http://bcb.unl.edu/dbCAN2/download/Databases/dbCAN-HMMdb-V6.txt] and CAZy database v07312018 [http://bcb.unl.edu/dbCAN2/download/CAZyDB.07312018.fa] were used for metagenome and metaproteome analyses.

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

## Acknowledgements

This work was funded by the Max Planck Society and supported by the Deutsche Forschungsgemeinschaft (DFG) grant HE 7217/1-1, and through the Cluster of Excellence "The Ocean Floor — Earth's Uncharted Interface" project 390741603 to J.-H.H.; B.M.F., H.T., D.Becher, T.S., R.A., and J.-H.H. received funding by the DFG research unit FOR2406 (POMPU). We thank the captain and crew of the research vessel Aade for provision of seawater samples; K. Wiltshire from the Alfred Wegener Institute in Helgoland for provision of physicochemical and algae biodiversity data; T. Harder from the University of Bremen for providing the diatom *C. socialis* strain isolated by his group at station Kabeltonne in the 2016 spring bloom; and O. Tranquet from the Institut National de la Recherche Agronomique in Nantes for providing the INRA antibodies. We thank P. L. Buttigieg for discussion on data; A. Ellrott for help with microscopy; L. Franzmeyer and K.-P. Rücknagel for DAPI counts; I. Ulber for DOC quantification; A. Bolte for HPAEC-PAD measurements; and K. Imhoff for sulphate quantification.

## Author contributions

S.V.-M. and J.-H.H. designed the study. S.V.-M. performed the polysaccharide extractions, microarray analyses, immunofluorescence microscopy, epitope detection chromatography and ELISA; A.S. performed monosaccharide analysis and statistical analyses;

J.N. executed DOC analysis; A.W. supported with TFF processing during the sampling campaign; T.B.F. performed metagenome analysis; D.Bartosik, D.Becher and T.S. performed metaproteome analysis. S.V.-M., A.S., T.B.F., D.B., J.N., A.W., W.G.T.W., B.M.F., H.T., D.B., T.S., R.A., and J.-H.H. discussed the results. S.V.-M. and J.-H.H. wrote the manuscript. All authors commented and approved the manuscript.

## Funding

## Competing interests

The authors declare no competing interests.
