## [Peer Review File · Nature Communications]

REVIEWER COMMENTS

Reviewer #1 (Remarks to the Author):

Review of MS entitled "Diatom fucan polysaccharide precipitates carbon during an algal bloom" by Silvia Vidal-Melgosa et al.

Analysing organic matter, or specifically polysaccharides in seawater has been a long standing challenge. This paper describes a new approach to the problem, and teases apart dynamics of different polysaccharides during diatom blooms. This investigation also likely provides a partial answer with regards to the composition of TEP (Transparent Exopolymer Particles): The dynamic of the FCSP between the dissolved and particulate pools, and in relation to diatom cells during the progression of a bloom are identical to those described for TEP. TEP created by phytoplankton, especially diatoms have also been found to be very sulfur-rich, similar to FCSP.

This is an exiting paper, especially because of the novel tools used to determine polysaccharide composition. This is also a detailed and well balanced study. It is worthy of publication in Nature Communications. Below some points to consider:

Loss process of organic matter during the sampling period include grazing and sinking, not only microbial degradation. Although, quite possibly an enrichment of FCSP is due to reduced microbial degradation, it could also be due to reduced stickiness and sinking or due to reduced palatability. This needs to be acknowledged. Actually, the paper implies that FCSP are both, accumulating due to low degradation AND sticking ("adhesive") which would mean it would sink out and not accumulate. This un-addressed inconsistency in the argument creates some tension in the story.

An important part of the story is the question if the HMWDOM contains only extracellular material, or if cell lysis during processing could have contributed to the HMWDOM. In fact it seems that analysed polysaccharides contain both, intra and extracellular compounds (exudates and cell wall material). What evidence is there that the FCSP are extracellular (exudates)? This is implied, but the distinction is central to the interpretation, and needs to be addressed outright.

The described method to measure total cell number (using DAPI on 0.2 μ m filters) enumerates bacteria, and micro-phytoplankton would not be representative in such samples. Please state clearly (both prokaryotes and eukaryotes are cells). One figure talks about total phytoplankton cells – likely shown are total micro-phytoplankton cells, ignoring pico-phytoplankton. How were diatoms and coccoids counted and identified (magnification, # enumerated, microscope type). Please check all the inconsistencies/ missing information with regards to cells.

Chaetoceros socialis is different than most other diatoms in that several chains are often clustered, connected by a central additional spine and enclosed in a transparent mucousy-gel matrix. These units are referred to as colonies (not aggregates as they form by growth not coagulation). This species is therefore well-known to generate large amounts of extracellular material that is very visible even with the naked eye and known to sediment (e.g. Booth et al 2002, DSR II, 49:5003-5025). Sampling and handling would destroy this mucus envelope/ colony and create "particle fragments". It is worthwhile to mention this.

It is stated that FCSP in HMWDOM decreased over time and increased in POM. How did the total concentration change? How were the dynamics between DOM and POM for the other polysaccharides?

The sampling period covered several successive al diatom blooms, so calling this a bloom period, or referring to "blooms" in plural would be more correct.

Line 168-171 – re write sentence

190 – TEP particles were – omit “particles” (TEP = transparent exopolymer particles)
203: Remove “Then”
219 & 223 - repetitive

Reviewer #2 (Remarks to the Author):

This excellent manuscript presents compelling data that identifies fucose-containing polysaccharides from diatoms as components of accumulating particulate matter that is an important aspect of carbon capture to ocean floors during algal blooms. This is exciting and of considerable interest as it opens up a previously undocumented part of the ocean carbon cycles and dynamics. It links fucans with diatoms for the first time.

The manuscript is excellent in terms of its writing style and structure – being comprehensive and systematic in its presentation. The manuscript outlines discoveries made with up to date methods including carbohydrate microarrays and these discoveries would unlikely to have been possible with alternative current methodologies. The data sets are generally very well presented and clear and will readily engage readers. Substantive and comprehensive extra data sets, that have provided the data included in the main paper, are included. The experimental work appears to have been done to a very high standard and the appropriately detailed methods can be readily followed.

The final discussion is clear and concise – perhaps too concise. Perhaps the authors could consider a little more the movement of fucans from dissolved organic matter to POM and how this relates to the polymer at diatom surfaces. Is it soluble, separate from diatom cells, at the start of the bloom? Does it then aggregate? Or is it mostly attached to diatom surfaces throughout – these issues could be clarified. What is meant by calling these polysaccharides adhesive – is this just part of the aggregation process?

In relation to this, during the analyses the HMWDOM is extracted with solvents EDTA and NaOH – there seems little logic to this as the material is already soluble. This does not detract from the observations but it may be worth noting.

The authors have used an impressive array of monoclonal antibodies to probe the microarrays but these are unlikely to cover all the possible polysaccharide structures found in ocean water and therefore line 186 may need a little refinement.

In Fig 2a the fluorescence signals are hard to disentangle and this may be a particular problem for readers with impaired colour vision. Some annotation of things would help here. What are the structures? An equivalent bright field image may help in this regard too.

Reviewer #3 (Remarks to the Author):

The manuscript “Diatom fucan polysaccharide precipitates carbon during an algal bloom” by Vidal-Melgosa et al., describes a discovery of a fucose-containing sulphated polysaccharide (FCSP) that resists degradation by bacterial enzymes in the ocean. It is known that algal polysaccharides may act as adhesives by binding together organic and inorganic molecules into larger particles, which can sink and sequester carbon into deeper marine waters. Thus a novel candidate compound that may act as a carbon sequestration agent in the ocean by using an unexplored mechanism for the stabilisation of organic matter in the ocean would present an exciting scientific discovery.

I appreciate a combination of chemical and meta -omics approaches that allows a comprehensive analysis of the chemical and microbial compositions of the seawater samples. However, I have some concerns that prevent me from recommending the current version of this manuscript for publication. In particular, I find that (i) the use of metagenomics isn’t clearly justified and its results are not presented in the most efficient way and (II) the discussion and conclusion sections of the manuscript do not provide enough substance to be published in Nature Communications.

Major comments:

1. Could you explain your methodological reasoning as to why metagenomics was used to look for genes coding for the enzymes involved in the degradation of the polysaccharides described in the manuscript? It appears from reading the manuscript that metagenomics results do not support the main conclusion, proposing FCSP as a novel polysaccharide resistant to biodegradation. I think that a more clear explanation is needed as to why this methodology has been chosen and the significance of its results should be presented in a more structured manner. As of now, I fail to see how metagenomics results add value to the conclusions made by the authors. Please see my thoughts on this issue in minor comments below.

2. I think that the Discussion section needs to be overall stronger. As of now, it compares FCSP to other polysaccharides providing important and often protective functions in other biological systems, including plant roots and animals. However, there's no discussion of one of the more interesting questions in my opinion that emerged from this study. I think the reader will want to see the discussion of the potential trade-off between the protective function of FCSP and their role as an adhesive that leads to particle formation and promotes sinking of diatom cells (lines 329-333).

Minor comments:

In my version of the manuscript the Introduction and Results sections are not labeled

Lines 62-64. Can they be partially degraded but still be able to sink attached to other particles? Can you provide a reference or data discussing the timeframe needed to maintain the integrity of particles? Are there data supporting the mentioned 10 days period?

Line 158. I'm not sure I would call it constant composition. There are some clear peaks suggesting that different taxa dominated the community at different stages.

Line 159. What supports this assumption? Since the algal composition has changed over time (according to fig 6a) the production intensity could also change. Please clarify.

Line 162. Can you define somewhere in the text what you mean by bacterial degradation in this case? Does it mean that this compound cannot be degraded at all, i.e. no chemical bonds can be broken and no changes can be made to its structure? Or does it mean that changes made by bacterial enzymes are not sufficient to break it down enough so that its properties would drastically change?

Line 266. Bacteria consume laminarin over FCSP

I think that this section would benefit if it was rewritten in a more unified manner following a common structure. Two types of compounds are described here. It will be easier for the reader to follow the results if they were described uniformly. Which bacterial groups are responsible for their degradation? What are the metagenomics results? What are the metaproteomics results? Not necessarily in this order but in the SAME order.

Lines 282-283. Can you discuss these results in more detail? Why do you think you have detected GH16 on all dates and it was highly expressed as opposed to GH3, GH30, and GH17? What's known about these enzymes in the literature, do all of them need to be active for a successful degradation of laminarin? From how the results are presented, it is rather difficult to appreciate their significance. Metagenomics detected genes coding for most of the enzymes required for both processes (laminarin and FCSP degradation). These results do not support the conclusion that FCSP is more stable compared to laminarin. Metaproteomics provides results in favor of this conclusion. But then again, what is the purpose of the metagenomics analysis?

Lines 296-299. I think that sulphated polysaccharides are more stable because their structures are more difficult to break down, not because they challenge microbes. I understand the idea behind this sentence but I don't think it's expressed in the best manner. Maybe rewrite the sentence?

Lines 299-302. It appears that the functional metaG data and metaP data do not support each other. I'm confused as to why metaG was used in these experiments in the first place (see next comment). The metaG taxonomic results demonstrate that Bacteroidia and Gammaproteobacteria encoded the enzymes required for laminarin degradation and that Verrucomicrobiae were the key degraders of FCSP, both facts known before as stated in lines 274-275 and 290.

Lines 300-301. Could you explain why the choice was made to use metagenomics instead of metatranscriptomics in concert with metaproteomics? Would access not just to the presence/absence of genes but also to their expression be more useful in this study?

Line 305. ... abundances of genes coding for the enzymes... Same comment to phrasing throughout the text. MetaG shows abundance of genes coding for enzymes.

Line 319. I think this is the first time in the manuscript where the algae type is specified. Why? What about other types of algae, do they all produce FCSP, same type, different? Since you mention the algae type, a brief explanation of its uniqueness (or not) would be appropriate.

Lines 320-321. Given what we know about their release and travel speed, how fast do they leave the phycosphere? What is the difference in the FCSP concentrations in the macroalgal phycosphere and in the open water? Have studies been done measuring their concentrations in a range of distances from the cell wall? These compounds may regulate the algal microbiome because of their antibacterial properties if their concentrations are high enough to actually deter bacterial cells. I think a more thorough discussion is needed in this section.

Lines 329-332. I would like to see a further discussion of this trade-off.

Lines 332-333. I think that how microbial interactions regulate carbon balance (not just export) is hardly an emerging question, it has been studied for quite some time on physiological, molecular, and -omics levels. I believe that including some of the latest findings from these studies would make the discussion section stronger.

Submission of the revised manuscript NCOMMS-20-33382-T

We thank the three reviewers for their time and effort during the evaluation of our manuscript. We are grateful to your comments and have given them careful consideration. We have made changes and additions based on your comments, which we believe have significantly improved the manuscript.

REVIEWER COMMENTS

Reviewer #1 (Remarks to the Author):

Review of MS entitled “Diatom fucan polysaccharide precipitates carbon during an algal bloom” by Silvia Vidal-Melgosa et al.

Analysing organic matter, or specifically polysaccharides in seawater has been a long standing challenge. This paper describes a new approach to the problem, and teases apart dynamics of different polysaccharides during diatom blooms. This investigation also likely provides a partial answer with regards to the composition of TEP (Transparent Exopolymer Particles): The dynamic of the FCSP between the dissolved and particulate pools, and in relation to diatom cells during the progression of a bloom are identical to those described for TEP. TEP created by phytoplankton, especially diatoms have also been found to be very sulfur-rich, similar to FCSP.

- This is an exiting paper, especially because of the novel tools used to determine polysaccharide composition. This is also a detailed and well balanced study. It is worthy of publication in Nature Communications. Below some points to consider:

We thank the reviewer for this positive evaluation.

- Loss process of organic matter during the sampling period include grazing and sinking, not only microbial degradation. Although, quite possibly an enrichment of FCSP is due to reduced microbial degradation, it could also be due to reduced stickiness and sinking or due to reduced palatability. This needs to be acknowledged. Actually, the paper implies that FCSP are both, accumulating due to low degradation AND sticking (“adhesive”) which would mean it would sink out and not accumulate. This un-addressed inconsistency in the argument creates some tension in the story.

Thanks for pointing this out, we agree and have now acknowledged the potential for reduced grazing by zooplankton (lines 185-187). Based on our data we consider FCSP is “sticky/adhesive” and contributes to particle formation. Particles would eventually sink resulting in a net FCSP removal. We address the “inconsistency” by stating in the revised manuscript: “Notably, we observed the highest increase of FCSP in the smallest particulate fraction, potentially because these smaller particles are neutrally buoyant and therefore less prone to removal by sinking” (lines 204-206).

We consider that the low degradation of FCSP allows its accumulation in POM, which includes the FCSP-containing particles > 0.2 µm (some will sink) but also FCSP that is coating the diatom cells. (Reflection added in lines 221-223).

- An important part of the story is the question if the HMWDOM contains only extracellular material, or if cell lysis during processing could have contributed to the HMWDOM. In fact it seems that analysed polysaccharides contain both, intra and extracellular compounds (exudates and cell wall material). What evidence is there that the FCSP are extracellular (exudates)? This is implied, but the distinction is central to the interpretation, and needs to be addressed outright.

We tested the presence of FCSP in diatom laboratory cultures (including different species) and found that FCSP is a diatom-secreted polysaccharide highly abundant in the dissolved exudate fraction. We separated the diatom cells from the dissolved fraction either by centrifugation (6800 x g) or by filtration (0.2 µm) at 200 mbar to avoid/minimize cell lysis and FCSP was confirmed to be a released exudate. We found FCSP coating the diatom cell and also released into the HMWDOM pool. A statement addressing this has been included in lines 243-246.

In regards to our field data, we cannot exclude the possibility that there was some cell lysis during filtration, which would result in polysaccharides that belonged to the diatom cell ending up in the HMWDOM. However, although the same sampling process was maintained during the 21 sampling days, the FCSP in HMWDOM and POM showed opposite trends. If cell lysis was an important contributor to our HMWDOM, its trend should be similar to the FCSP-containing matter present in POM (which would “leak” FCSP). Therefore, this suggests that the vast majority of FCSP in our HMWDOM samples originates from microalgae released exudates.

- The described method to measure total cell number (using DAPI on 0.2 um filters) enumerates bacteria, and micro-phytoplankton would not be representative in such samples. Please state clearly (both prokaryotes and eukaryotes are cells).

Microbial cells were enumerated via an automated image acquisition and analysis system as described in (Bennke, et al. 2016. “Modification of a high-throughput automatic microbial cell enumeration system for shipboard analyses.” <https://doi.org/10.1128/AEM.03931-15>), this reference has now been included in the Methods section. In the used settings, we count everything stained between 25 and 250 pixels in size at a given fluorescence signal above background. The individual pixel size is 0.1016 µm, which corresponds to cell sizes between approx. 0.5 - 5 µm. We cannot exclude that at some time points small picoeukaryotes contribute in the low percentage range, as well and especially towards summer there are up to 10% Archaea present at Helgoland. Hence, we stay with the term but have added “microbial” - Total microbial cell numbers.

- One figure talks about total phytoplankton cells – likely shown are total micro-phytoplankton cells, ignoring pico-phytoplankton. How were diatoms and coccos counted and identified (magnification, # enumerated, microscope type). Please check all the inconsistencies/ missing information with regards to cells.

Yes, as you pointed out pico-phytoplankton were not counted but the nano- and micro-phytoplankton cells were counted.

The phytoplankton cell counts were performed as part of the long-term monitoring program at Helgoland Roads, where since 1962 the phytoplankton species composition has been monitored. We have now included the information of how they were specifically counted and identified in the corresponding Methods section.

At Helgoland Roads LTER the counts are reported as “phytoplankton cell counts/total abundance of phytoplankton/phytoplankton composition” (e.g. Wiltshire et al. 2008, <https://doi.org/10.4319/lo.2008.53.4.1294>; Wiltshire et al. 2004, <https://doi.org/10.1007/s10152-004-0192-4>). Therefore we would like to keep this term. But we have now added “phytoplankton (including nano-phytoplankton and micro-phytoplankton)” to the Supplementary Figure legend as well as to the Methods section to make it clear that pico-phytoplankton were not counted.

- Chaetoceros socialis is different than most other diatoms in that several chains are often clustered, connected by a central additional spine and enclosed in a transparent mucousy-gel matrix. These units are referred to as colonies (not aggregates as they form by growth not coagulation). This species is therefore well-known to generate large amounts of extracellular material that is very visible even with the naked eye and known to sediment (e.g. Booth et al 2002, DSR II, 49:5003-5025). Sampling and handling would destroy this mucus envelope/ colony and create “particle fragments”. It is worthwhile to mention this.

We thank the reviewer for pointing this out. To address this point we have included this sentence: “Some of the particles may derive from disrupted C. socialis colonies, which contain clusters of diatom chains covered in mucous that can fragment during the sampling process.” (lines 223-225)

- It is stated that FCSP in HMWDOM decreased over time and increased in POM. How did the total concentration change? How were the dynamics between DOM and POM for the other polysaccharides?

With the here used approach we could show that concentrations increase or decrease but we cannot yet provide absolute concentrations. For this one would need an enzymatic approach as recently shown for laminarin (Becker et al. 2020). At the moment such a method to quantify FCSP does not exist. We show the dynamics of other polysaccharides between DOM and POM in Figure 1 and in the corresponding complete Supplementary Fig. 5. Other polysaccharides increased in POM but they did not decrease in HMWDOM at the same time. FCSP was the only one showing dynamics that indicate aggregation as previously described for TEP. In an effort to provide a number for FCSP we used

quantitative ELISA and determined there was a 7-fold decrease in HMWDOM and a 3-fold increase in POM FCSP concentration during the bloom. However, as mentioned in Supplementary Discussion in the section “Quantitative ELISA to investigate FCSP fold change in HMWDOM and POM”, for the analysis we used a macroalgal fucan as standard, but absolute numbers may vary in dependence of the fucan standard used, thus not providing absolute concentrations.

- The sampling period covered several successive diatom blooms, so calling this a bloom period, or referring to “blooms” in plural would be more correct.

We have included “blooms” or “bloom period” whenever suitable.

- Line 168-171 – re write sentence

Done

- 190 – TEP particles were – omit “particles” (TEP = transparent exopolymer particles)

Done

- 203: Remove “Then”

Done

- 219 & 223 - repetitive

The repetition has been removed.

Reviewer #2 (Remarks to the Author):

This excellent manuscript presents compelling data that identifies fucose-containing polysaccharides from diatoms as components of accumulating particulate matter that is an important aspect of carbon capture to ocean floors during algal blooms. This is exciting and of considerable interest as it opens up a previously undocumented part of the ocean carbon cycles and dynamics. It links fucans with diatoms for the first time.

The manuscript is excellent in terms of its writing style and structure – being comprehensive and

systematic in its presentation. The manuscript outlines discoveries made with up to date methods including carbohydrate microarrays and these discoveries would unlikely to have been possible with alternative current methodologies. The data sets are generally very well presented and clear and will readily engage readers. Substantive and comprehensive extra data sets, that have provided the data included in the main paper, are included. The experimental work appears to have been done to a very high standard and the appropriately detailed methods can be readily followed.

We thank the reviewer for this very positive and encouraging evaluation.

- The final discussion is clear and concise – perhaps too concise. Perhaps the authors could consider a little more the movement of fucans from dissolved organic matter to POM and how this relates to the polymer at diatom surfaces. Is it soluble, separate from diatom cells, at the start of the bloom? Does it then aggregate? Or is it mostly attached to diatom surfaces throughout – these issues could be clarified.

We thank the reviewer for these suggestions and agree, the discussion was too concise. We discuss these suggested points now in the new longer discussion in the revised version of the manuscript.

- What is meant by calling these polysaccharides adhesive – is this just part of the aggregation process?

By adhesive we mean that the polysaccharide has physicochemical features that enhance its assembly properties and lead to formation of TEP-like particles (and subsequently can act as a glue between diatom cells and other molecules promoting aggregation). We have now included a statement to clarify what we mean by adhesive (lines 384-386).

- In relation to this, during the analyses the HMWDOM is extracted with solvents EDTA and NaOH – there seems little logic to this as the material is already soluble. This does not detract from the observations but it may be worth noting.

We agree that this is worth mentioning, thanks for pointing this out. We have added our reasoning for that in the text section where polysaccharides' solubility and extractability are discussed (Supplementary Discussion lines 143-149).

- The authors have used an impressive array of monoclonal antibodies to probe the microarrays but these are unlikely to cover all the possible polysaccharide structures found in ocean water and therefore line 186 may need a little refinement.

The sentence was revised. The sentence now clarifies that we only refer to the other polysaccharides that were detected in this study and not to all possible structures present in the ocean.

- In Fig 2a the fluorescence signals are hard to disentangle and this may be a particular problem for readers with impaired colour vision. Some annotation of things would help here. What are the structures? An equivalent bright field image may help in this regard too.

Thanks for these suggestions, we agree and have added white arrows and arrowheads to highlight key features adding this information to the figure legend. An equivalent bright field image has as well been added in this regard.

Reviewer #3 (Remarks to the Author):

The manuscript “Diatom fucan polysaccharide precipitates carbon during an algal bloom” by Vidal-Melgosa et al., describes a discovery of a fucose-containing sulphated polysaccharide (FCSP) that resists degradation by bacterial enzymes in the ocean. It is known that algal polysaccharides may act as adhesives by binding together organic and inorganic molecules into larger particles, which can sink and sequester carbon into deeper marine waters. Thus a novel candidate compound that may act as a carbon sequestration agent in the ocean by using an unexplored mechanism for the stabilisation of organic matter in the ocean would present an exciting scientific discovery.

I appreciate a combination of chemical and meta -omics approaches that allows a comprehensive analysis of the chemical and microbial compositions of the seawater samples. However, I have some concerns that prevent me from recommending the current version of this manuscript for publication. In particular, I find that (i) the use of metagenomics isn't clearly justified and its results are not presented in the most efficient way and (II) the discussion and conclusion sections of the manuscript do not provide enough substance to be published in Nature Communications.

We thank the reviewer for these positive, critical comments. We have expanded the metagenomics and metaproteomics section, explaining in more detail why we chose these methods and presenting the results in an organized way. We have also extensively expanded the discussion and added a paragraph about the suggested trade-off.

Major comments:

- 1. Could you explain your methodological reasoning as to why metagenomics was used to look for genes coding for the enzymes involved in the degradation of the polysaccharides described in the manuscript? It appears from reading the manuscript that metagenomics results do not support the main conclusion, proposing FCSP as a novel polysaccharide resistant to biodegradation. I think that a more clear explanation is needed as to why this methodology has been chosen and the significance of its

results should be presented in a more structured manner. As of now, I fail to see how metagenomics results add value to the conclusions made by the authors. Please see my thoughts on this issue in minor comments below.

We have added detail to the metagenomics and metaproteomics section explaining why we chose these methods and why we focused on the presented genes/proteins. We have also ordered this section, as suggested, to better organize the results.

- 2. I think that the Discussion section needs to be overall stronger. As of now, it compares FCSP to other polysaccharides providing important and often protective functions in other biological systems, including plant roots and animals. However, there's no discussion of one of the more interesting questions in my opinion that emerged from this study. I think the reader will want to see the discussion of the potential trade-off between the protective function of FCSP and their role as an adhesive that leads to particle formation and promotes sinking of diatom cells (lines 329-333).

We have now extensively expanded the discussion by writing more about the trade-off as well as the secretion and aggregation of the FCSP in this context.

Minor comments:

- In my version of the manuscript the Introduction and Results sections are not labeled

We included the missing Introduction and Results headers.

- Lines 62-64. Can they be partially degraded but still be able to sink attached to other particles? Can you provide a reference or data discussing the timeframe needed to maintain the integrity of particles? Are there data supporting the mentioned 10 days period?

If the backbone of the polysaccharide is degraded by endo-enzymes its solubility will drastically increase. The smaller the fragments the more soluble they are. Such a backbone cutting polysaccharide cannot hold the particle together anymore. Maybe the smaller polysaccharide fragments can still stick to particles but they cannot serve as glue. This can be observed by adding enzymes to gel forming polysaccharides such as alginate, agar, carrageenan (Hehemann et al. 2012). Adding enzymes dissolves the gel particles.

We provide several references for the particle sinking rate 100m/day. Our 10 day estimate is based on our theoretical consideration that we describe in this paper. The depth at which carbon can be stored long term is often described as below 1000 m. A particle that sinks 100m/day takes 10 days to reach the carbon storage depth of 1000 m. So the polysaccharide or polysaccharides that hold a particle together must be stable for at least 10 days. This is just a theoretical back on the envelope calculation

to illustrate the problem to the reader and we think it works quite well so we would like to keep it as it is.

- Line 158. I'm not sure I would call it constant composition. There are some clear peaks suggesting that different taxa dominated the community at different stages.

The sentence was changed accordingly.

- Line 159. What supports this assumption? Since the algal composition has changed over time (according to fig 6a) the production intensity could also change. Please clarify.

The previous sentence has now been changed to make clear that diatoms were always present and may have continuously produced polysaccharides. We have also qualified this statement by preceding it with "Assuming".

- Line 162. Can you define somewhere in the text what you mean by bacterial degradation in this case? Does it mean that this compound cannot be degraded at all, i.e. no chemical bonds can be broken and no changes can be made to its structure? Or does it mean that changes made by bacterial enzymes are not sufficient to break it down enough so that its properties would drastically change?

In the revised version this has been defined (lines 165-171). Degradation of the polysaccharide backbone by endo-enzymes would lead to rapid dissolution. Modifications of the polysaccharide by other enzymes (e.g. sulphatases) may also alter solubility but to a lesser extent.

- Line 266. Bacteria consume laminarin over FCSP

I think that this section would benefit if it was rewritten in a more unified manner following a common structure. Two types of compounds are described here. It will be easier for the reader to follow the results if they were described uniformly. Which bacterial groups are responsible for their degradation? What are the metagenomics results? What are the metaproteomics results? Not necessarily in this order but in the SAME order.

Thank you for this very helpful comment, the section was rewritten with the suggested order.

- Lines 282-283. Can you discuss these results in more detail? Why do you think you have detected GH16 on all dates and it was highly expressed as opposed to GH3, GH30, and GH17? What's known about these enzymes in the literature, do all of them need to be active for a successful degradation of laminarin? From how the results are presented, it is rather difficult to appreciate their significance.

Metagenomics detected genes coding for most of the enzymes required for both processes (laminarin and FCSP degradation). These results do not support the conclusion that FCSP is more stable compared to laminarin. Metaproteomics provides results in favor of this conclusion. But then again, what is the purpose of the metagenomics analysis?

We have expanded on the explanation of what is known about GH16, GH17, GH30 and GH3. We also included a statement about their requirement for laminarin degradation (lines 323-326), i.e. the whole repertoire may probably allow to obtain most glucose from laminarin but GH16 allows to rapidly start its degradation. We think that this (together with what we now state in lines 312-314) may be the reason why GH16 was highly expressed during the diatom blooms, which has also been found in previous studies (Teeling et al., 2012).

We have added context for the choice of metagenomics, to identify known degraders of these polysaccharides and their potential for polysaccharide degradation (genes). We also explain that metaproteomics was chosen to identify if those genes were expressed and we focused especially on known endo-acting and exo-acting enzymes that would hydrolyze the laminarin and FCSP. We think that the fact that the metagenomics shows that enzyme genes for the polysaccharide degradation are present but that they appear not to be used (expressed) or are at a concentration too low to effect substantial FCSP removal is an important result to the reader.

- Lines 296-299. I think that sulphated polysaccharides are more stable because their structures are more difficult to break down, not because they challenge microbes. I understand the idea behind this sentence but I don't think it's expressed in the best manner. Maybe rewrite the sentence?

The sentence has been rewritten.

- Lines 299-302. It appears that the functional metaG data and metaP data do not support each other. I'm confused as to why metaG was used in these experiments in the first place (see next comment). The metaG taxonomic results demonstrate that Bacteroidia and Gammaproteobacteria encoded the enzymes required for laminarin degradation and that Verrucomicrobiae were the key degraders of FCSP, both facts known before as stated in lines 274-275 and 290.

Yes, that certain members of the Gammaproteobacteria and Bacteroidia are laminarin degraders and that Verrucomicrobiae are fucan degraders was known from previous publications. But that these bacteria were present during this bloom and that their polysaccharide degrading genes were present during this bloom was not known. Therefore, we used metagenomics to verify if polysaccharide degraders and their enzyme genes were present in this bloom. Next we asked if they also used these genes and made the enzymes and tested this by using proteomics.

- Lines 300-301. Could you explain why the choice was made to use metagenomics instead of metatranscriptomics in concert with metaproteomics? Would access not just to the presence/absence of genes but also to their expression be more useful in this study?

This was a deliberate choice. Although a combination of metatranscriptomics and metaproteomics would be valuable, our filtration approach for sampling takes normally about four hours and therefore there might be issues with mRNA stability. Provided that we had access to a world class proteomics facility we opted for expression profiling with proteomics. Moreover, the detection of enzymes with proteomics is more direct given as the expression level of mRNA does not always scale with protein expression and amount. Furthermore, we aimed at detecting the genetic potential and tracing the genetic potential for polysaccharide degradation in stable DNA cannot be substituted by metatranscriptomics, which can see what is expressed but not genes that are present and not expressed.

- Line 305. ... abundances of genes coding for the enzymes... Same comment to phrasing throughout the text. MetaG shows abundance of genes coding for enzymes.

Thanks for pointing this inaccuracy out. This is now correctly defined in the figure legends and throughout the manuscript.

- Line 319. I think this is the first time in the manuscript where the algae type is specified. Why? What about other types of algae, do they all produce FCSP, same type, different? Since you mention the algae type, a brief explanation of its uniqueness (or not) would be appropriate.

FCSP (types defined in lines 183-185) is so far only known to be produced by brown macroalgae, some tunicates and here we show diatoms such as the *Chaetoceros* spp. also produce them. In the new extended discussion, we have provided more detail about which organisms make FCSP (lines 428-434).

- Lines 320-321. Given what we know about their release and travel speed, how fast do they leave the phycosphere? What is the difference in the FCSP concentrations in the macroalgal phycosphere and in the open water? Have studies been done measuring their concentrations in a range of distances from the cell wall? These compounds may regulate the algal microbiome because of their antibacterial properties if their concentrations are high enough to actually deter bacterial cells. I think a more thorough discussion is needed in this section.

No studies have been carried out regarding FCSP concentration in distance of or close to the cell wall of brown macroalgae. To achieve quantitative measurements of FCSP in distance to diatoms or macroalgae one would need tools to quantify specifically one polysaccharide structure but these are only now being developed (Becker et al. 2020). The situation for FCSP on the surface of brown algae is similar. Since there are no studies about the quantitative influence on microbiome or phycosphere

biogeochemistry, it is unfortunately not possible to adequately discuss this interesting issue. That FCSP might play a role in modulating the macroalgal microbiome is still a supposition stemming from laboratory cultures where purified FCSPs show antimicrobial activity.

- Lines 329-332. I would like to see a further discussion of this trade-off.

We have expanded the discussion about this trade-off and further covered FCSP secretion and particle formation.

- Lines 332-333. I think that how microbial interactions regulate carbon balance (not just export) is hardly an emerging question, it has been studied for quite some time on physiological, molecular, and -omics levels. I believe that including some of the latest findings from these studies would make the discussion section stronger.

We agree in that microbial interactions that regulate carbon balance have been widely studied. We most likely did not define it well in our previous 332-333 statement, but we were referring to the microbial interactions that shape the evolution of the FCSP-diatom trade-off. Our question was, how microbial interactions promote/regulate the diatom secretion of FCSP to the exterior of their cells? We have now expanded this part to make it clear that we refer to the defence-aggregation trade-off and how microbial interactions may regulate it (lines 453-459), which would influence the number of particles and ultimately the organic matter export.

REVIEWERS' COMMENTS

Reviewer #1 (Remarks to the Author):

The paper has greatly improved. Just a couple of comments and typos.

It is of course correct that phytoplankton needs light to photosynthesize, but since the cited Smetacek 1985 paper it has become well accepted that aggregation and sinking is an important part of the lifecycle of most diatom species including Chaetoceros and Thalassiosira. Most (or all) diatoms survive better in the dark (cold and high nutrient concentrations) than in light when macro nutrients are limiting. This makes aggregation and sinking out of the euphotic zone at the end of blooms an important survival strategy for a species and gives it the opportunity to develop “seed cells”. Additionally, aggregates provide a microenvironment (high nutrient recycling) allowing some fraction of the enclosed diatom cells to form spores (at the cost of other cells within the aggregate that are “recycled” or by using the mentioned metals like Fe, that are enriched in the exudates). Leaving the euphotic zone due to aggregation is thus commonly not considered a disadvantage for diatoms. The listed possible benefits of antiviral and bacterial protection additional help build a seeding population, especially as the danger of infection would be large in the high concentration environment of an aggregate. Much of this overall picture can not be easily and clearly proven in a scientific sense, but a large amount of individual observations support this overall view as aggregation being an important part of the diatom lifecycle. The authors may thus consider reformulating a couple of their statements (e.g. lines 387, 389, 405, 410, 414) to reflect this.

I also want to add a personal comment. Since Zhou et al found that TEP are rich in polysaccharides enriched in fucose and sulfate half esters over 20 years ago, the question on the composition of TEP has not moved significantly forward. I really enjoyed the novel approach of this study and am extremely happy to see progress being made.

Line 99 remove “a”

Line 267 ...and increaseds

Line 268 – why are their 3 images under (a) and (b) respectively

Line 272 (a, b)

Line 276/277 – why are there two figures in (e)?

Line 279 ..same for the five shown... – which five?

Line 281 – the second column in each case seems to show sulfate, but in the legend sulfate is listed directly in line with all the monomers. Please separate so the graph can be understood more easily.

Reviewer #2 (Remarks to the Author):

The authors have responded appropriately to the comments and revised the manuscript accordingly.

Reviewer #3 (Remarks to the Author):

I think that the manuscript has improved significantly by the revision process. The current version is easy to follow and the information is presented in a more straightforward way, thanks to the many changes made to the text. The discussion section has been significantly expanded and it now provides interesting ideas regarding the FCSP secretion and particle formation, and how microbial interactions may regulate the defence-aggregation trade-off. The authors have addressed all of my comments.

I think that the current version of the manuscript can be recommended for publication.

Minor comments:

Lines 81-84. I think it would be better to keep the wording consistent here. MetaP and metaG showed a high abundance/expression of enzymes for the degradation of labile polysaccharides... and low abundance/expression of enzymes for the degradation of FCSP...

Lines 202-203. Found that FCSP was abundant

Lines 340-341 ...the metagenome/metagenomic data showed that Verrucomicrobiae...

Submission of the revised manuscript NCOMMS-20-33382-A

We thank the editor and the three reviewers for their evaluation of our revised manuscript.

REVIEWERS' COMMENTS

Reviewer #1 (Remarks to the Author):

The paper has greatly improved. Just a couple of comments and typos.

- It is of course correct that phytoplankton needs light to photosynthesize, but since the cited Smetacek 1985 paper it has become well accepted that aggregation and sinking is an important part of the lifecycle of most diatom species including *Chaetoceros* and *Thalassiosira*. Most (or all) diatoms survive better in the dark (cold and high nutrient concentrations) than in light when macro nutrients are limiting. This makes aggregation and sinking out of the euphotic zone at the end of blooms an important survival strategy for a species and gives it the opportunity to develop “seed cells”. Additionally, aggregates provide a microenvironment (high nutrient recycling) allowing some fraction of the enclosed diatom cells to form spores (at the cost of other cells within the aggregate that are “recycled” or by using the mentioned metals like Fe, that are enriched in the exudates). Leaving the euphotic zone due to aggregation is thus commonly not considered a disadvantage for diatoms. The listed possible benefits of antiviral and bacterial protection additional help build a seeding population, especially as the danger of infection would be large in the high concentration environment of an aggregate. Much of this overall picture can not be easily and clearly proven in a scientific sense, but a large amount of individual observations support this overall view as aggregation being an important part of the diatom lifecycle. The authors may thus consider reformulating a couple of their statements (e.g. lines 387, 389, 405, 410, 414) to reflect this.

We thank the reviewer for this very valuable comment. We have taken these points into consideration and have rephrased and modified some sentences of the discussion. We now state in the discussion that aggregation and sinking at the end of blooms is a natural part of the lifecycle of most planktonic diatoms (resting stage cells can remain at lower depths or in the sediment during non-favourable conditions). At the same time, we discuss that the observed high secretion of FCSP at the very beginning of the bloom seems detrimental, as soluble FCSP promotes formation of particles that most likely contribute to cell sinking, while during early phases diatoms would rather need to stay at the surface to be able to successfully bloom during favourable conditions. In addition, although FCSP cell coating was highest at the end of the bloom (which could be due to cells “promoting their own sinking” by making themselves more sticky and prone to aggregate on purpose because of conditions turning unfavourable), FCSP was also coating the cells already at the beginning of the bloom. We propose that the observed secretion of FCSP since already early phases might be due to the mentioned defence-aggregation tradeoff.

- I also want to add a personal comment. Since Zhou et al found that TEP are rich in polysaccharides enriched in fucose and sulfate half esters over 20 years ago, the question on the composition of TEP has not moved significantly forward. I really enjoyed the novel approach of this study and am extremely happy to see progress being made.

Thank you for these encouraging words.

- Line 99 remove “a”

Done

- Line 267 ...and increaseds

Done

- Line 268 – why are their 3 images under (a) and (b) respectively

The immunolabelling microscopy experiments were performed four times independently. In all the cases we observed an increase of FCSP coating the diatom cells and appearance of FCSP-containing particles in the filters from the end of the bloom. Each of the panels presents three representative images from the beginning and from the end, because we want to illustrate with some examples the shape and size of the FCSP-particles detected at the end of the bloom as well as the lack of FCSP-particles at the beginning illustrated by the black background.

- Line 272 (a, b)

“In a-b” has been added. The (b) is referring to the end of the bloom panel and now after the dot “In a-b” has been included.

- Line 276/277 – why are there two figures in (e)?

The three figures in panels (e) and (f) show the result of applying the same approach to separate FCSP from crude polysaccharide extracts, i.e. separation by chromatography followed by detection with ELISA. We included two figures in panel (e) because they are connected as they correspond to biomass from the bloom. Therefore, panel (e) as a whole shows that FCSP was present in both POM and HMWDOM in both beginning and end of the bloom. In contrast, panel (f) is totally different as it corresponds to biomass from a lab culture. If possible we would like to keep it like this.

- Line 279 ..same for the five shown... – which five?

We have now better defined the five single anion exchange chromatography runs and have also included (four in e and one in f) to make clear which ones we are referring to.

- Line 281 – the second column in each case seems to show sulfate, but in the legend sulfate is listed directly in line with all the monomers. Please separate so the graph can be understood more easily.

Thanks for pointing this out. We agree and have now separated the sulfate from the monosaccharides in the legend to make the figure easier to understand.

Reviewer #2 (Remarks to the Author):

- The authors have responded appropriately to the comments and revised the manuscript accordingly.

We thank the reviewer for this positive evaluation.

Reviewer #3 (Remarks to the Author):

- I think that the manuscript has improved significantly by the revision process. The current version is easy to follow and the information is presented in a more straightforward way, thanks to the many changes made to the text. The discussion section has been significantly expanded and it now provides interesting ideas regarding the FCSP secretion and particle formation, and how microbial interactions may regulate the defence-aggregation trade-off. The authors have addressed all of my comments.

I think that the current version of the manuscript can be recommended for publication.

We thank the reviewer for the very positive evaluation of the changes we made and for supporting our manuscript for publication.

Minor comments:

- Lines 81-84. I think it would be better to keep the wording consistent here. MetaP and metaG showed a high abundance/expression of enzymes for the degradation of labile polysaccharides... and low abundance/expression of enzymes for the degradation of FCSP...

Thanks for this suggestion. We agree, it is better to keep the wording consistent when comparing these findings. This has been addressed.

- Lines 202-203. Found that FCSP was abundant

The word “that” has been added.

- Lines 340-341 ...the metagenome/metagenomic data showed that Verrucomicrobiae...

We have added “that”.